# Study of the thermal regime of a reservoir on the Qinghai-Tibetan Plateau, China

**Yanjing Yang, Yun Deng** ⓘ *, **Youcai Tuo, Jia Li, Tianfu He, Min Chen**

State Key Laboratory of Hydraulics and Mountain River Engineering, Sichuan University, Chengdu, China

* dengyun@scu.edu.cn

**Data Availability Statement:** All relevant data are within the paper and its Supporting Information files.

**Funding:** This work was supported by the National Key R&D Program of China (grant

## Abstract

The Qinghai-Tibetan Plateau region has unique meteorological characteristics, with low air temperature, low air pressure, low humidity, little precipitation, and strong diurnal variation. A two-dimensional hydrodynamic CE-QUAL-W2 model was configured for the Pangduo Reservoir to better understand the thermal structure and diurnal variation inside the reservoir under the local climate and hydrological conditions on the Qinghai-Tibetan Plateau. Observation data were used to verify the model, and the results showed that the average error of the 6 profile measured monthly from August to December 2016 was 0.1˚C, and the root-mean-square error (RMSE) was 0.173˚C. The water temperature from August 2016 to September 2017 was simulated by inputting measured data as model inputs. The results revealed that the reservoir of the Qinghai-Tibetan Plateau was a typical dimictic reservoir and the water mixed vertically at the end of March and the end of October. During the heating period, thermal stratification occurred, with strong diurnal variation in the epilimnion. The mean variance of the diurnal water temperature was 0.10 within a 5 m water depth but 0.04 in the whole water column. The mixing mode of inflow changed from undercurrent, horizontal-invaded flow and surface layer flow in one day. In winter, the diurnal variation was weak due to the thermal protection of the ice cover, while the mean variance of diurnal water temperature was 0.00 within both 5 m and the whole water column. Compared to reservoirs in areas with low altitude but the same latitude, significant differences occurred between the temperature structure of the low-altitude reservoir and the Pangduo Reservoir (P<0.01). The Pangduo Reservoir presented a shorter stratification period and weaker stratification stability, and the annual average $SI$ value was 26.4 kg/m$^2$, which was only 7.5% that of the low-altitude reservoir. The seasonal changes in the net heat flux received by the surface layers determined the seasonal cycle of stratification and mixing in reservoirs. This study provided a scientific understanding of the thermal changes in stratified reservoirs under the special geographical and meteorological conditions on the Qinghai-Tibetan Plateau. Moreover, this model can serve as a reference for adaptive management of similar dimictic reservoirs in cold and high-altitude areas.

number2016YFC0502202) and the National Natural Science Foundation of China (grant numbers 91547211).

**Competing interests:** The authors have declared that no competing interests exist.

## 1. Introduction

When a river dam is used to regulate runoff, it will inevitably affect the spatial and temporal changes in water temperature in the reservoir area and downstream rivers [1]. Thermal pollution of rivers degrades water quality, increases nutrients and threatens ecosystem health [2–4]. The velocity of the shallow channel is fast, and the vertical mixing is basically uniform. However, the construction of the dam transforms the water into a relatively static or slow-flowing general state, thus forming unique reservoir temperature stratification characteristics [5, 6]. The stratified thermal structure is the basic physical characteristic of the lake, which increases the risk of water quality deterioration, such as the reduction of nutrients and dissolved oxygen in deep water [7–9]. Moreover, the temperature stratification of reservoir water will affect the utilization of reservoir water, and this process likely has a negative influence on reservoir and river aquatic life in water ecosystems [10–12]. Research on the evolution law of the water thermal regime is the basis for optimizing the allocation of reservoir water and mitigating its negative effects [13, 14]. Researchers in many countries have studied the water temperature of reservoirs based on their specific conditions [15–18].

Surface mixed layers of reservoirs periodically appear and disappear with variations in heat flux [19]. The heat absorption of the water surface has an important influence on the heating process of the thermocline during the heating period. In the cooling period, the heat loss of the surface water drives vertical mixing and the inflow of low-temperature water forms an intrusion flow, which jointly controls the cooling process of the thermocline [20, 21]. Temperature, radiation and local winds are the most important factors in the thermal processes in lakes or reservoirs [22–24]. However, the impact of climate change is unique because the stratified characteristics of lakes (reservoirs) are affected by various factors, such as the density structures, morphological characteristics (deep run-of-the-river and lake-type reservoirs), and artificial regulation of reservoirs [25–28]. Therefore, it is meaningful to explore the response to climate change and the thermal state of the reservoir.

In recent years, a set of highly indicative description systems was established [29]. The stratification time and the thermocline characteristics are important indexes to evaluate the response of lake thermal stratification to external factors [30]. The current research mainly focuses on the regularity of water temperature stratification in response to annual and seasonal changes [31, 32]. However, related research indicates that the reservoir would have a more subtle mixing process within a day; specifically, the epilimnion is heated during the daytime and cooled at night and subject to the influence of temperature fluctuations every day and night, resulting in diurnal variations [33]. Surface water always has a noticeable convection effect, which likely weakens the buoyancy effect of surface water [34]. The stratification of lakes and reservoirs is related to local meteorology, water temperature and geographical morphology [8]. The diurnal variation characteristics of lakes in different regions are different. In temperate reservoirs, the solar radiation reaching the surface is sufficient to produce a density difference and strong stability in the water column while the nighttime cooling of the surface water is not sufficient to break the buoyancy and form a heat cycle [35]. The density difference between the epilimnion and hypolimnion of tropical monomictic reservoirs is smaller than that of temperate reservoirs; therefore, tropical monomictic reservoirs are not only unstable but also prone to diurnal variations [36]. Dimictic reservoirs are covered with ice during part of the year and stratified during the other part, with two mixed periods in between. Few studies have focused on the diurnal variation characteristics and formation mechanisms of dimictic reservoirs in cold and high-altitude areas.

With the development of China's water conservancy projects in the source region of the Qinghai-Tibetan Plateau, the evolution of lake and reservoir water temperature is becoming

increasingly complicated [24]. The Qinghai-Tibetan Plateau has unique weather and atmospheric circulation characteristics, which have a significant impact on the Asian monsoon, global atmospheric circulation and global climate change [37, 38]. The Qinghai-Tibetan Plateau is characterized by low temperature, low air pressure, low humidity, little precipitation, a simple ecosystem structure and weak anti-interference ability. In addition, the Qinghai-Tibetan Plateau is ecologically vulnerable to human activities and environmental changes [39–41]. Local aquatic organisms on the Qinghai-Tibetan Plateau grow slowly and are more sensitive to changes in hydraulic conditions, such as water temperature and flow velocity. Thus, the impact of water temperature changes on the ecological environment will be more significant [42]. Few studies have investigated the water temperature characteristics of reservoirs on the Qinghai-Tibetan Plateau. Previous studies have demonstrated that the epilimnion, metalimnion and hypolimnion layers could be obviously distinguished in Nam Co based on their different physicochemical features [43]. Moreover, the thermal characteristics of deep run-of-the-river reservoirs on the Qinghai-Tibetan Plateau are rarely understood, and the mechanism of water mixing in reservoirs is still unclear.

Various numerical models have been developed and applied to study the hydrodynamic and thermal processes of reservoirs [44]. An unsteady two-dimensional model (along longitudinal and vertical directions) is the minimum requirement to study spatial-temporal thermal regime variations in the Pangduo Reservoir (Fig 1), which is a long-but-narrow reservoir with a longitudinal gradient in temperature that is not small enough to be neglected. Consequently, we conducted the present study by using CE-QUAL-W2, a two-dimensional laterally averaged hydrodynamic and water quality model [45]. This model provides accurate simulations of the stratification and density currents and has been widely used to study various reservoirs and river impoundments worldwide [46–48].

The present study focuses on the Pangduo Reservoir, a run-of-river reservoir located on the Qinghai-Tibetan Plateau in China. The study is conducted to better understand the thermal structure and diurnal variations inside the reservoir under the local climate and hydrological conditions on the Qinghai-Tibetan Plateau and identify the dominant factors that control the changes in diurnal variations. We investigated the changes of thermal regime in the Pangduo Reservoir by analyzing observed data in several monitored stations and by using the two-dimensional CE-QUAL-W2 model. Furthermore, the validated model was developed to understand the thermal structure and diurnal variations in the Pangduo Reservoir under the influence of local climate hydrological conditions on the Qinghai-Tibetan Plateau. In addition, a comparative case study was performed with a low-altitude reservoir.

## 2. Case study

### 2.1 Study area

The Qinghai-Tibetan Plateau is the highest geomorphic tectonic unit in the world and ranges from N 26˚00' to N 39˚47' and E 73˚19' to E 104˚47'. A unique hydrothermal condition has been formed due to the restriction of atmospheric circulation and plateau topography. The Pangduo Reservoir (N 30˚51'; E 91˚21') is located in the upper reaches of the Lhasa River, a tributary of Yarlung Zangbo River, and it is the largest water conservancy project on the Qinghai-Tibet Plateau. The Pangduo Reservoir consists of an embankment dam, flood releasing structures, and power tunnels that lead water to huge underground powerhouses. The top elevation of the dam is 4100 m above sea level (a.s.l.), and the elevation of the riverbed near the dam is 4038 m a.s.l. In the Pangduo dam, there are two 7-m high and 6-m wide power tunnels, with the entrance bottom elevation at 4059 m a.s.l. The basin of the Pangduo Reservoir has a typical canyon shape, and the slope of the riverbed is approximately 0.24%. The length of the

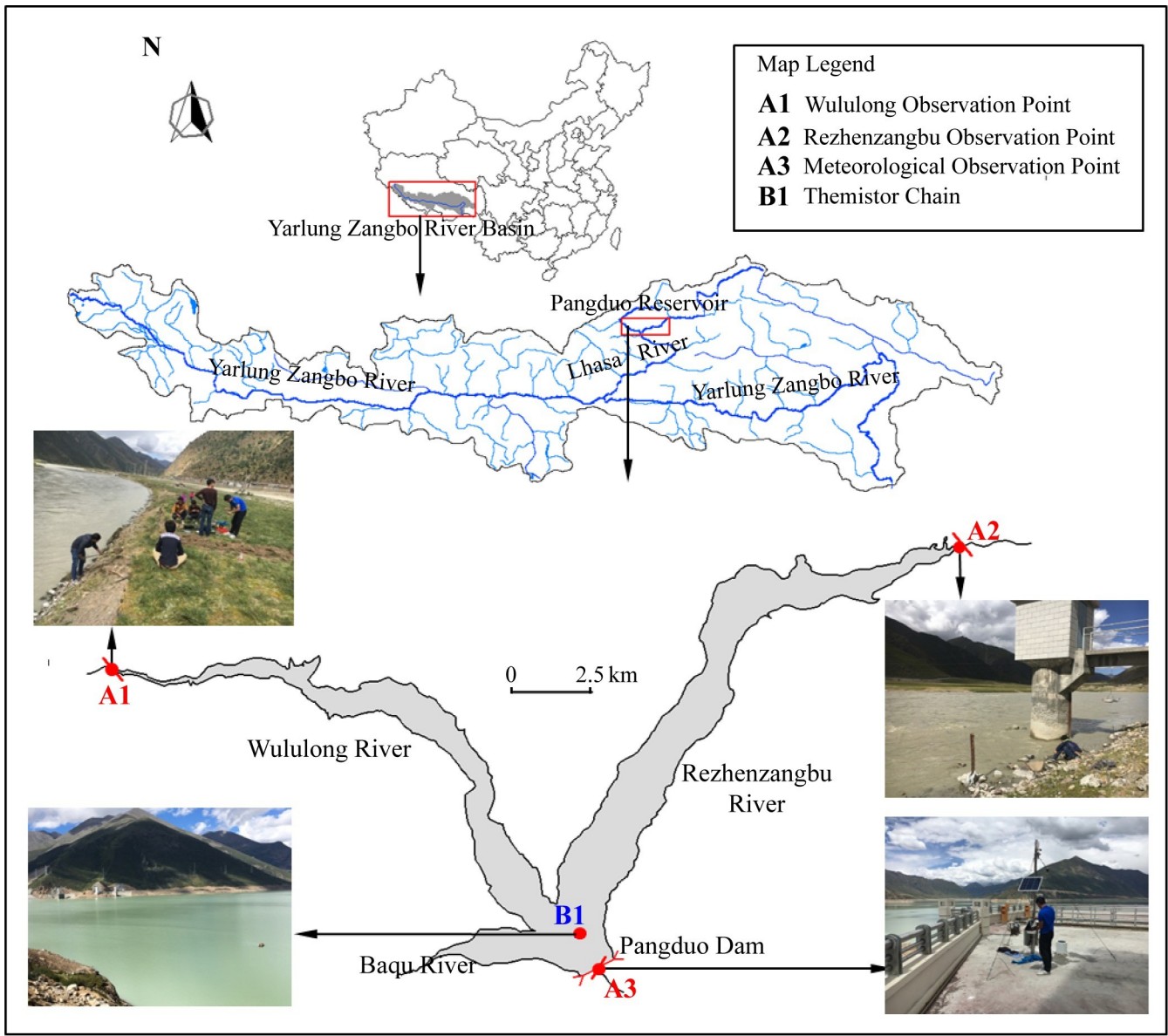

**Fig 1. Study area diagram.**

reservoir is approximately 46.51 km where the water surface is near horizontal and surface width varies from 100 m to 1700 m. The water depth near the dam ranges from 42 to 57 m throughout the year. The full storage of the reservoir is $12.3 \times 10^8$ m$^3$ at 4095 m a.s.l. The installed capacity of the power station is 120 MW. The function of the Pangduo Reservoir is irrigation and power generation, and the reservoir also provides functions that include flood control, urban water supply, and ecological environmental protection.

## 2.2 Regional monitoring

To understand the thermal characteristics of this plateau reservoir and its relationship with meteorological responses, we monitored the water temperature and meteorological conditions of the Pangduo Reservoir from August 23, 2016 to September 28, 2017. The monitoring

**Table 1. Monitoring scheme.**

| Project | Point Location | Latitude and Longitude | Instrument | Period and Frequency |
|---|---|---|---|---|
| Continuous water temperature | A1: Wululong | 91˚12'19"E30˚15'51"N | ZDR-20h | 2016.08.25~2017.09.28, Once per hour. |
| | A2: Rezhenzangbu | 91˚31'17"E30˚18'20"N | | |
| Meteorological | A3: Dam top | 91˚21'49"E30˚10'55"N | Jinzhou SunshinePC-4 | |
| Vertical water temperature | B1: Upstream of dam | 91˚21'09"E30˚11'15"N | TidbiT v2 (UTBI-001) | 2016.08.25~2016.08.31, Once per hour. |
| Manual monitoring | B1: Upstream of dam | 91˚21'09"E30˚11'15"N | EXO2 | 2016.09~2016.12, Once a month |

scheme is shown in Table 1. The monitoring points (Fig 1A1 and 1A2) were set up at the end of the Wululong and Rezhenzangbu tributary reservoirs, and the monitoring frequency was once per hour. A vertical water temperature automatic monitoring temperature chain was installed in front of the dam for short-term monitoring (Fig 1B1) from August 25 to August 31, 2016; the monitoring frequency was once per hour; and the vertical interval of sampling depth was 1 m. Manual on-site inspections and reviews were conducted monthly on September 15, October 15, November 22 and December 15, during which we used EXO2 to monitor the vertical water temperature. Other factors, such as air temperature, ambient humidity, wind speed, wind direction, and solar radiation, were monitored by a self-recording weather station (Fig 1A3) installed on the top of the dam during the monitoring period.

The average monthly air temperature varied from -2.1˚C to 13.7˚C (S1A Fig), with the lowest air temperature in January and the highest air temperature in July, with an average variance of 11.2˚C in one day. The lowest air temperature appeared from 6 to 7 a.m., and the mean was 2.1˚C. The highest temperature was from 3 to 4 p.m., and the mean was 11.0˚C. The average solar radiation was 231.2 W/m$^2$ during the monitoring period (S1B Fig), which showed an upward trend from December to August of the following year and reached the highest value in August at 359 W/m$^2$. The solar radiation was the strongest during the day from 12 to 2 p.m., with an average value of 980.4 W/m$^2$. The mean water temperatures of the Rezhenzangbu (S1C Fig) and Wululong branches (S1D Fig) were 6.1˚C and 6.7˚C, respectively, and they were affected by meteorological conditions. The average daily variations in the temperature of the Rezhenzangbu tributary and Wululong tributary were 2.1˚C and 3.2˚C, respectively.

## 2.3 Satellite pictures in winter

The Pangduo Reservoir freezes in winter. To understand the growth, development, and ablation processes of winter ice in the Pangduo Reservoir, we used Landsat 8 satellite data to identify the classification of ice and water in the reservoir area based on the spectral differences of different features (S2 Fig). With the decrease in air and inflow water temperature in early December and affected by the water-gas heat exchange and the solid boundary heat conduction, ice first appeared in the shallow water of the reservoir tail, with ice coverage of 6%. In mid-February, as the water body continued to lose heat, the initial ice cover formed in the Wululong branch, the water surface was frozen, and the ice area of the Rezhenzangbu branch increased rapidly. The area in front of the dam was frozen, and the ice coverage rate in the reservoir area reached 67.35%. As the air and inflow water temperature increased by the end of March, the ice surface began to thaw and the ice coverage rate in the reservoir area declined to 50.4%. Finally, the freezing period ended at the end of April.

## 3. Mathematical models and methods

Our research activities do not require specific permissions, the research area is open, there are no restrictions on scientific research, and the research does not involve endangered or protected

species. We used CE-QUAL-W2, a two-dimensional hydrodynamics and water quality model [45], to quantify the thermal regime and diurnal variation of the reservoirs on the Qinghai-Tibetan Plateau. This paper constructed a longitudinal and vertical two-dimensional water temperature model for the whole reservoir area, and it ignored the variation in various variables along the width of the river; thus, the model could be applied to water bodies with longitudinal and vertical temperature gradients in the Pangduo Reservoir. The main input boundaries of the model include inflow, outflow, inflow water temperature, and meteorological conditions.

## 3.1 Governing equations

The governing equations [45] include mass conservation and conservation of momentum using the Boussinesq approximation, in which the effects of density changes are considered only in the gravity term and the hydrostatic pressure assumption is used. The specific details are shown in Table 2.

## 3.2 Model grid and calculation conditions

The Pangduo Reservoir was divided into $131 \times 37$ (longitudinal × vertical) rectangular cells. The longitudinal size of the cell grid was $100 \sim 300$ m, and the vertical size was 2 m (S3 Fig). The verification period was from August 25, 2016 to December 31, 2016. The temperature field at the initial moment was obtained after interpolation based on the vertical water temperature measured in the reservoir area on August 25, 2016. The inflow water temperature and meteorological data were based on measured data. The water level, outflow and inflow were based on measured data from the Pangduo Hydrological Station. The calculation period was from August 25, 2016 to August 24, 2017. The inflow water temperature and meteorological data were based on measured data. The reservoir operation data adopted the designed operating conditions of the Pangduo Hydropower Station (S4 Fig).

To more clearly compare the thermal conditions between the reservoir on the Qinghai-Tibetan Plateau and those in the low- and medium-altitude regions, the same mathematical model was used to replace the water temperature and meteorological boundary of the reservoirs in the low-altitude area with an average altitude of 180 m at the same latitude (S5 Fig). During the calculation period, the average air temperature and inflow water temperature were higher (20.9°C and 19.9°C, respectively) and the solar radiation was lower (118.79 W/m$^2$).

## 3.3 Thermal stratification evaluation index

We selected indicators to quantify the thermal stratification of water bodies and evaluate the characteristics of the stratified structure of the water temperature. The vertical water temperature gradient ($VTG$, °C/m) [49] and the buoyancy frequency ($N$, 1/s) [50] were calculated based on the water temperature calculation results of the vertical section in front of the dam.

$$VTG = \frac{\partial T(z)}{\partial z} \tag{1}$$

$$N = \sqrt{-\frac{g}{\rho_0}\frac{\partial \rho(z)}{\partial z}} \tag{2}$$

where $T(z)$ is the water temperature at depth z, °C; $\rho(z)$ is the density at depth z, kg/m$^3$, which is temperature dependent [45]; $\rho_0$ is the average density of the whole water column, kg/m$^3$; $g$ is the acceleration of gravity, m/s$^2$; and $N$ is the important indicator used in limnology and oceanography and manmade reservoirs. The change trend of $N$ can be used to evaluate the

**Table 2. Governing equations.**

| Field | Name | Equations | Variates and coefficients |
|---|---|---|---|
| 1 | Continuous equation | $\frac{\partial BU}{\partial x} + \frac{\partial BW}{\partial z} = (q + q_b)B$ | $B$: width of the water body |
| | | | $U$: longitudinal flow velocity |
| | | | $W$: vertical flow velocity |
| | | | $q$: lateral unit length of the incoming flow |
| 2 | X-direction momentum equation | $\frac{\partial UB}{\partial t} + \frac{\partial UUB}{\partial x} + \frac{\partial WUB}{\partial z} = gB\sin\alpha + g\cos\alpha B\frac{\partial \eta}{\partial x}$ $-\frac{g\cos\alpha B}{\rho}\int_\eta^z \frac{\partial \rho}{\partial x}dz + \frac{1}{\rho}\frac{\partial B\tau_{xx}}{\partial x} + \frac{1}{\rho}\frac{\partial B\tau_{xz}}{\partial z} + q_b BU_x$ | $\eta$: water level |
| | | | $\alpha$: river dip |
| | | | $\rho$: water density |
| | | | $U_x$: $x$ component of the tributary flow rate |
| | | | $U_b$: longitudinal flow velocity of the tributary |
| | | | $B$: angle of the dry tributary |
| | | | $q_b$: unit length of the tributary |
| | | $U_x = U_b\cos\beta$ | $B_\eta$: water surface width |
| 3 | Hydrostatic pressure assumption | $0 = g\cos\alpha - \frac{1}{\rho}\frac{\partial p}{\partial z}$ | $D_x$ and $D_z$: longitudinal and vertical dispersion coefficients |
| | | | $S_\Phi$: laterally averaged source/sink term |
| 4 | Free surface equation | $B_\eta \frac{\partial \eta}{\partial t} = \frac{\partial}{\partial x}\int_\eta^h Budz - \int_\eta^h qBdz$ | |
| 5 | Heat transport equation | $\frac{\partial B\rho C_p T}{\partial t} + \frac{\partial UB\rho C_p T}{\partial x} + \frac{\partial WB\rho C_p T}{\partial z}$ $-\frac{\partial\left(BD_x\frac{\partial \rho C_p T}{\partial x}\right)}{\partial x} - \frac{\partial\left(BD_z\frac{\partial \rho C_p T}{\partial z}\right)}{\partial z} = S_\Phi B$ | |
| 6 | Water-air interface heat exchange | $\varphi_{GY} = \varphi_{sn} + \varphi_{an} - \varphi_{br} - \varphi_\varepsilon + \varphi_c$ | $\varphi_{GY}$: plateau water-air heat exchange flux |
| | | | $\varphi_{sn}$: solar shortwave radiant heat flux |
| | | | $\varphi_{an}$: atmospheric longwave radiant heat flux |
| | | | $\varphi_{br}$: back radiant heat flux by water |
| | | | $\varphi_\varepsilon$: the evaporative heat flux |
| | | | $\varphi_c$: heat conduction flux |
| 7 | Ice cover | $\rho_i L_f \frac{\Delta h}{\Delta t} = h_{ai}(T_i - T_e) - h_{wi}(T_w - T_m)$ | $\rho_i$: ice density |
| | | | $L_f$: latent heat of freezing |
| | | | $\Delta h/\Delta t$: ice thickness growth rate |
| | | | $h_{ai}$ and $h_{wi}$: heat exchange coefficients of ice-gas and ice-water, respectively |
| | | | $T_m$: temperature of the ice-water interface |
| | | | $T_e$: temperature at which the heat exchange between ice and air reaches equilibrium |
| | | | $T_w$: water temperature under ice |
| | | | $T_i$: ice temperature |

stratified stability of water columns. $N$ indicates that in a stable temperature layered structure, the fluid particles move in the vertical direction after being disturbed. The combined effect of gravity and buoyancy always returns these factors to an equilibrium position, which oscillates due to inertia. The oscillation frequency can be understood as the exchange rate of water [50].

To better evaluate the changes in the stratification stability of the water column, the *SI* index, which reflects the stability of the deep water body, was used to evaluate the strength of the stratification stability of the reservoir [30].

$$SI = \int_{Z_0}^{Z_l} (Z - \bar{Z})\rho_Z dz \tag{3}$$

where $Z$ is the depth of the water column from the surface; $Z_0$, $Z_l$ and $\bar{Z}$ are the depths of the surface water, the lower end of the water column, and the centroid of the water column, respectively; and $\rho_z$ is the water density at depth Z. *SI* can be converted into energy by multiplying by the acceleration of gravity and the volume of each layer, and it represents the ideal energy estimate of the entire water column to achieve mixing in the depth range without an increase or decrease in heat.

Water age (days) is defined as the persistence of water entering the reservoir from upstream and describes the duration of time that water remains in a water body. In the CE-QUAL-W2 model, we set the zero-order decay rate to –1 per day and zeroing out all other generic constituent kinetic parameters results in a state variable that increases by 1 per day, which provides an exact representation of the water age or hydraulic residence time [45].

### 3.4 Statistics and analysis

Microsoft Excel version 2010 was used for all the statistical analyses. We used the average absolute error, root-mean-square error (RMSE) and standard deviation (STD) to evaluate the goodness of the fitted temperature curve, as well as the differences between observed temperatures and simulated temperatures. A paired T-test was conducted to determine the significance levels of the differences of the (a) water temperature structure, (b) stratification stability index, and (c) buoyancy frequency between the low-altitude reservoir and the Pangduo Reservoir. A value of $P<0.01$ was reported as significant.

## 4. Results and discussion

### 4.1 Monitoring data analysis

Strong diurnal variation occurred in the vertical water temperature distribution within a range of 10–20 m from the surface upstream of the dam from August 25 to August 31, 2016 (S6 Fig). The typical study layers include the surface layer and layers at 1, 2, 3, 4, 5, 10, 20, and 30 m below the surface. The daily variation in water temperature is shown in Table 3. The diurnal amplitude gradually decreased as the depth increased. The average daily variation in the surface layer over the 7 days was 4.1°C, with a range of 2.3°C ~ 5.7°C. The average daily variation was 0.7°C at a depth of 5 m below the surface layer, and the variation range was 0.4°C ~ 1.2°C. The average daily variation was 0.3°C at a depth of 30 m, which was only approximately 7.3% the value of the surface layer.

### 4.2 Model calibration and validation

The calibration of the Pangduo Reservoir's CE-QUAL-W2 model focused on several critical coefficients that had the greatest influence on the simulated temperature profiles upstream of

**Table 3. Water temperature variation.**

| Date | Air temperature variation/°C | Inflow temperature variation/°C | Water temperature variation/°C | | | | | | | | |
|---|---|---|---|---|---|---|---|---|---|---|---|
| | | | 0 | 1 | 2 | 3 | 4 | 5 | 10 | 20 | 30 |
| 2016/8/25 | 12.3 | 4.8 | 5.7 | 2.6 | 0.9 | 0.9 | 0.8 | 0.8 | 0.7 | 0.5 | 0.3 |
| 2016/8/26 | 9.6 | 4.5 | 3.9 | 1.5 | 1.2 | 1.0 | 0.6 | 0.5 | 1.1 | 0.3 | 0.3 |
| 2016/8/27 | 13.4 | 5.5 | 5.5 | 1.4 | 0.9 | 0.7 | 0.5 | 0.4 | 0.6 | 0.5 | 0.4 |
| 2016/8/28 | 12.6 | 4.9 | 5.7 | 3.0 | 2.5 | 1.3 | 0.8 | 0.5 | 0.4 | 0.4 | 0.4 |
| 2016/8/29 | 10.3 | 5.1 | 3.4 | 1.4 | 1.3 | 1.3 | 1.0 | 0.9 | 1.1 | 0.4 | 0.4 |
| 2016/8/30 | 6.8 | 5.2 | 2.3 | 0.9 | 0.9 | 1.1 | 1.2 | 1.2 | 0.5 | 0.4 | 0.3 |
| 2016/8/31 | 10.5 | 4.7 | 2.6 | 1.1 | 1.0 | 1.2 | 1.3 | 0.8 | 0.5 | 0.3 | 0.3 |
| Average | 10.8 | 5.0 | 4.1 | 1.7 | 1.3 | 1.1 | 0.9 | 0.7 | 0.7 | 0.4 | 0.3 |

the dam. The critical model parameters are the shading coefficients (ratio that allows incident shortwave solar radiation to reach the water surface in different segments due to topographic shelter), the wind sheltering coefficient (when multiplied by the wind speed, this coefficient reduces effects of the wind to take into account differences in the terrain between the measured station and the prototype site). The sensitivity analysis indicated that values of 0.8 for the shading coefficient (S7A Fig) and 1.0 for the wind sheltering coefficient (S7B Fig) were able to generate simulated temperature profiles with minimal discrepancies from the measured profiles.

We compared the calculated results and measured data for EXO2 upstream of the dam each month from August to December 2016 (S8 Fig). The validation period included the high temperature period in summer and the low temperature period before freezing. The results showed that the surface water temperature decreased from 14.5°C in late August to 4.6°C in December. The temperature variation trends of the epilimnion and metalimnion were consistent, and the temperature of the hypolimnion was generally well matched. The average error during the entire verification period was 0.1°C, the STD was 0.517°C, and the RMSE was 0.173°C. The scatter plot of the measured and calculated water temperature showed that the degree of dispersion of the error was small (S9 Fig). The determination coefficient between measured value and calculated value was 0.98. The results showed that the model can accurately simulate the influence of buoyancy flow and atmospheric heat exchange on the stratified structure of water temperature in the reservoir.

The ice variation was also considered in the calculations. The calculated results showed that the ice upstream of the dam began to appear in late December, and the maximum ice thickness appeared in mid-January, with a thickness of 0.35 m. The ice cover melted in late April (S10 Fig). The results showed that the growth, development and ablation of ice were in good agreement with the interpretation time obtained from the satellite images (S2 Fig).

## 4.3 Thermal structure characteristics

Fig 2 shows the two-dimensional distribution process of the water temperature (A), temperature gradient (B), buoyancy frequency (C), and water age (D) in front of the dam from August 25, 2016, to August 24, 2017. The results showed that the Pangduo Reservoir was a typical dimictic reservoir. The reservoir has two turnovers when the temperature differences between the surface and the bottom reach 0°C: one at the end of March and one at the end of October in autumn. After each major turnover, the reservoir area will maintain the same vertical temperature (4°C) for a period of time.

From September to November 2016, the reservoir was in a cooling period. The surface water temperature decreased from 14.7°C to 7.2°C, and the vertical temperature difference of the reservoir decreased from 4.0°C to 0.2°C. The position of the thermocline mainly existed within 20 m of the surface layer, and the vertical temperature gradient (VTG) changed within 0.3°C/m, with an average of 0.08°C/m. The buoyancy frequency varied between 0.1 $H_z$ and 0.6 $H_z$ and presented a continuous unstable state, which was believed to be a diurnal thermocline phenomenon caused by the special meteorological and incoming water temperature changes on the Qinghai-Tibetan Plateau. The water body in the reservoir was mainly replaced in the upper and middle layers. Its water age varied between 20 and 60 days. The retention time of the water at the bottom of the reservoir was relatively long at approximately 140 ~ 200 days; however, as the temperature of the inflow water decreased and the flow increased during the cooling period, vertical convection increased. At the end of October, the old water at the bottom of the reservoir was replaced and lifted.

Beginning in early December, the average daily temperature in the Pangduo area changed to a negative value and the water temperature in the reservoir began trending to a vertical

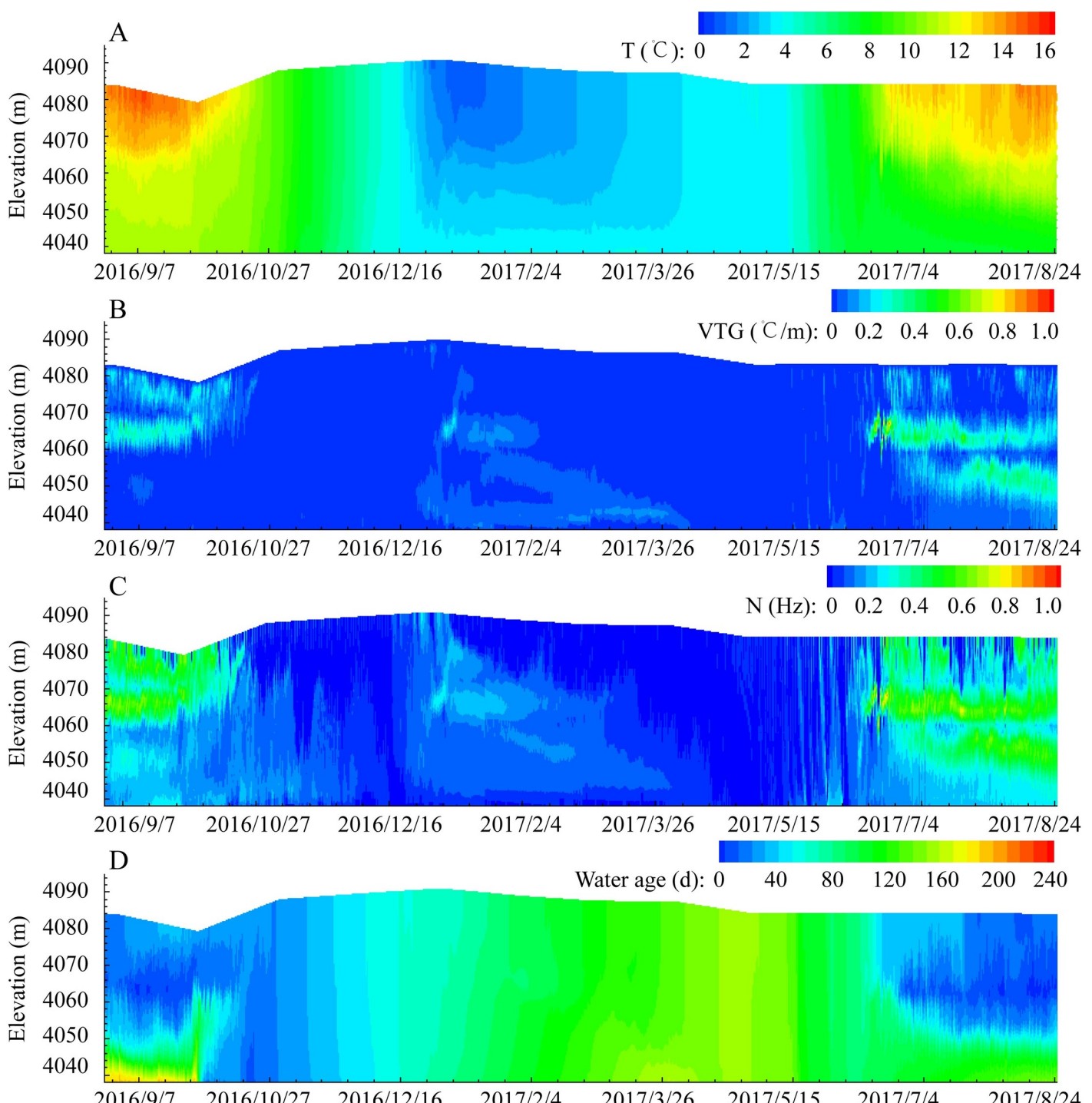

**Fig 2.** Vertical water temperature (A), gradient (B), buoyancy frequency (C), and water age (D) distribution in the calculation period of the Pangduo Reservoir.

mixing until the temperature decreased to 4˚C. The water surface temperature below 4˚C would soon drop to the freezing point to form the ice cover. When the surface temperature satisfied the condition of freezing, a static ice cover formed. The existence of ice caps transformed the heat exchange mode into the heat exchange between the surface water body and the

bottom of the ice cap and the heat exchange between the atmosphere and the upper surface of the ice cap [51]. An inverted structure of the vertical water body appeared. The temperature of the surface water body of the reservoir was approximately 0.5°C, and the temperature of the bottom of the reservoir was maintained at 4.0°C. Near the intake, the buoyancy frequency varied from 0.2 s⁻¹ to 0.3 s⁻¹. With the decrease in the flow, the retention of the water body increased and advanced evenly to the front of the dam between 120 and 180 days.

In May 2017, as the air and inflow water temperature increased, the temperature of the surface water also increased. The water body with a certain thickness of the surface layer began to convect and then reached the same temperature as that of the whole reservoir at approximately 4.0°C. By the end of May, the surface water temperature continued to increase to 7.4°C. Water temperature stratification began to occur in the reservoir from June to August. The surface water temperature increased to 13.8°C by the end of August, and the vertical temperature difference expanded to 6.5°C. At this time, the buoyancy frequency variation range of the surface layer in the range of 20 m expanded from 0.1 s⁻¹ to 0.6 s⁻¹ and a diurnal thermocline appeared. With the increase in the flow and the buoyancy, the replacement of the surface water body began to increase, changing between 20 and 60 days. The water temperature at the bottom of the reservoir was low and the water density was high, and the mix and exchange of water with the upper water body was hindered. The water age began to increase up to 160 days. Compared with the retention time of warm monomictic reservoirs and lakes, the retention time was relatively short [52, 53]. Due to the weak stratification of the reservoir area and the existence of a diurnal thermocline in Pangduo Reservoir, the replacement of the upper and middle water bodies occurred sooner than that in warm monomictic reservoirs [35].

## 4.4 Diurnal characteristics analysis

To explore the characteristics of diurnal variations in reservoir water temperature during the high-temperature period, the simulation results on August 15, 2017 were selected as a representative day. On August 15, the air temperature fluctuated between 10.3°C and 20.7°C, with the lowest temperature at 7 a.m. and the highest temperature at 4 p.m. The simulated results showed that the characteristics of the diurnal changes upstream of the dam were strong. The mean variance of diurnal water temperature in the water column was 0.04, while it was 0.10 within 5 m. The surface water temperature ranged from 14.0°C to 15.1°C, with the buoyancy frequency varying from 0.0 s⁻¹ to 0.6 s⁻¹. Meanwhile, the diurnal range of water temperature at the depth of 20 m was 12.1°C to 12.7°C. The vertical temperature difference from the surface to 20 m was 1.2°C on average, with a range from 1.1°C to 2.2°C. This phenomenon occurred because the average net heat flux (Fig 3A) was -164.17 W/m² during the period from 12 a.m. to 8 a.m. The surface water column lost heat and cooled, the density increased, the water column gradually mixed downward, and the water exchange rate increased. From 9 a.m. to 7 p.m., the average net heat flux of the surface water layer was 280.76 W/m². The surface water column absorbed heat and warmed, and the buoyancy of surface water increased, which inhibited the vertical mixing of the water column. Only the diffusion effect was used to transfer heat to the lower water layers, and there was a strong vertical temperature gradient until the temperature differences reached its peak at 7 p.m. From 7 p.m. to 11 p.m., the average net heat flux of the surface water body was -129.00 W/m² and the water body began to lose heat. This intraday thermal expansion and contraction will cause the mechanical disintegration of unstable structures due to gravity, leaving the surface water body in an unstable motion state [34].

The diurnal variation in the tail section of the reservoir was strong. The inflow changed over the entire depth range, and the daily temperature differences at the surface layer and the bottom layer were both 1.1°C, which was very unique. The mixing mode was mainly

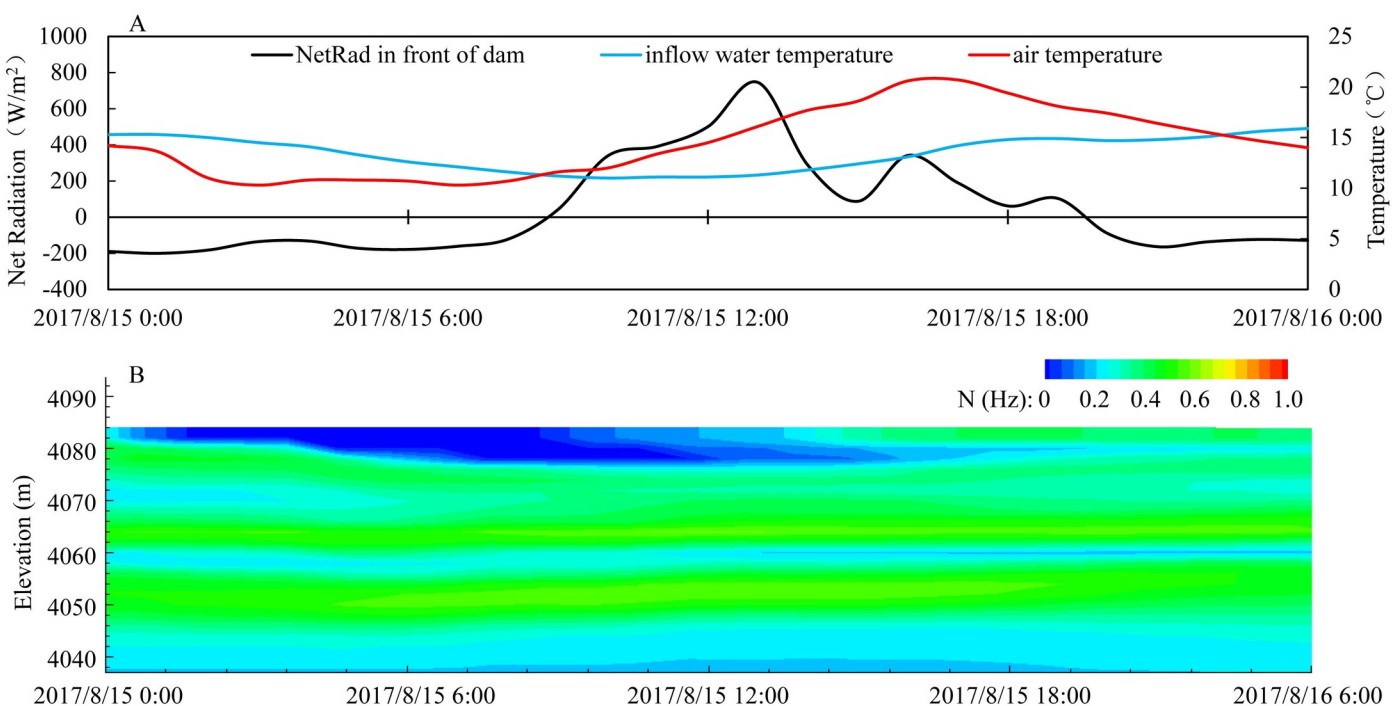

**Fig 3.** Diurnal distribution of net heat flux in front of the dam and inflow temperature (A) and buoyancy frequency in front of the dam (B) on August 15, 2017.

controlled by the influent water temperature change (Fig 3A). The temperature of the inflow water fluctuated between 11.0˚C and 15.6˚C, with a daily change of 4.6˚C. From 12 a.m. to 8 a. m., the average inflow water temperature was 13.8˚C, and the inflow water invaded the reservoir horizontally. From 9 a.m. to 7 p.m., the average water temperature of the inflow affected by glacial recharge decreased to 12.4˚C, the density of the water body increased, and a subsurface current gradually formed. From 8 p.m. to 11 p.m., the temperature of the inflow water increased, with an average value of 15.1˚C. The density of the inflow water decreased, leaving the river bottom and transitioning to the surface. When the water flowed into the mid-reservoir, it was affected by the temperature stratified anisotropic buoyancy flow and the inflow water temperature no longer dominated. The inflow mixing mode experienced undercurrent horizontal-invaded flow and surface layer flow in one day (S1 File). However, for most warm monomictic reservoirs, the inflow temperature into the reservoir was stable and the surface water temperature was usually higher than the inflow water temperature. At this time, the inflow water body would dive into the layer of the same density and had a stable dive line [49].

The diurnal variation was weak in winter (S11 Fig). The variance within 5 m and the whole water column were both 0.00. The reservoir area was completely frozen in February, and the ice layer provided thermal protection for the water body such that the heat exchange method of the surface water layer changed from "water-air heat exchange" to "ice-water heat exchange" [54, 55]. The inflow water at the tail of the reservoir was close to 0˚C. The water temperature structure of the Pangduo Reservoir remained inverted throughout the ice-sealing period with small differences.

## 4.5 Comparison with the low-altitude reservoirs at the same latitude

We selected August and December 2016 and March, June and September 2017 as the representative months, including the summer high temperature period, winter low temperature period

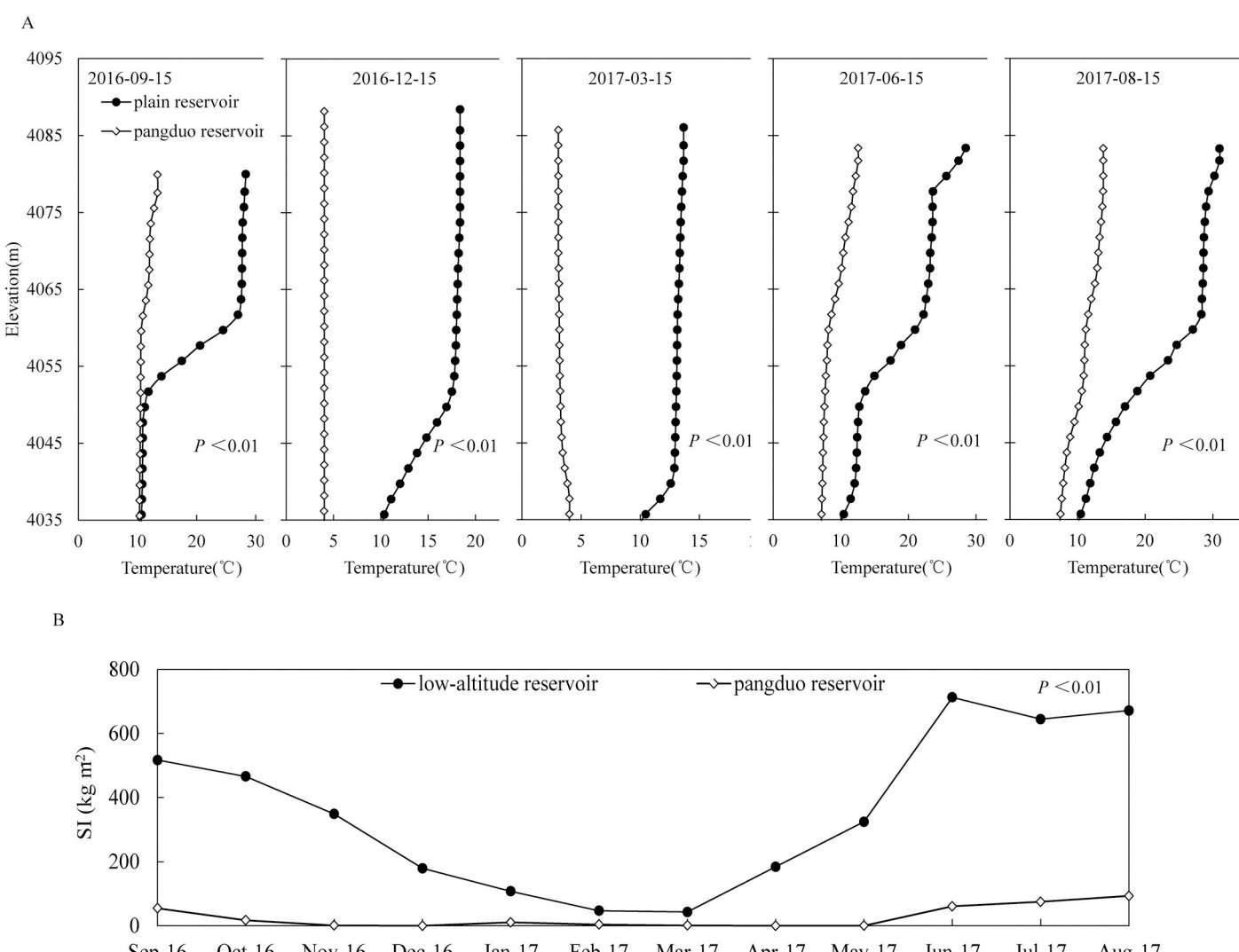

**Fig 4.** Comparison of the vertical water temperature between the low-altitude reservoir and the Pangduo Reservoir in front of the dam during the calculation period (A). Comparison of SI changes between the Pangduo Reservoir and the low-altitude reservoir (B). The *P* value is the result of the paired T test.

and spring heating process. There were significant differences between the temperature structure of the low-altitude reservoir and the Pangduo Reservoir (*P*<0.01). The water temperature was always greater than 4°C of the low-altitude reservoir, indicating that it is a typical warm monomictic reservoir (Fig 4A). From September to December 2016, the surface temperature decreased from 28.3°C to 17.0°C, with the vertical temperature difference narrowing from 17.8°C to 8.6°C. By the winter of February 2017, the surface water temperature had decreased to 13.8°C and the vertical temperature difference had narrowed to 4.3°C. Then, the surface water temperature began to rise to 31.0°C by August, with the vertical temperature difference increasing to 21.6°C.

During the warming period, the reservoir formed stable density stratification. Similar to most middle-low-altitude stratified reservoirs, the three-layer structure of the epilimnion, metalimnion and hypolimnion was maintained in front of the dam. The continuous decrease in air temperature during the cooling period cooled the surface water temperature, and the density of the surface water body increased. The sinking convection effect gradually deepened

from the surface layer first. In winter, the water column was vertically flipped and mixed within a depth of 40 m in front of the dam. Due to the deep depth, the water body at the bottom of the reservoir was not completely replaced [56, 57].

Fig 4B shows a comparison of the *SI* changes between the Pangduo Reservoir and the low-altitude reservoir monthly. A bigger *SI* indicates that the greater energy required to achieve vertical mixing of the water column, represents the more stable stratification. Significant differences occurred between the *SI* of the low-altitude reservoir and the Pangduo Reservoir ($P<0.01$). The average annual *SI* of the low-altitude reservoir was 353.75 kg/m$^2$ while that of the Pangduo Reservoir was 26.4 kg/m$^2$, which was 7.46% that of the low-altitude reservoir. The annual variation range of the low-altitude reservoir was 43.03~712.68 kg/m$^2$ while that of the Pangduo Reservoir was 0 ~ 93.13 kg/m$^2$, which was 13.91% that of the low-altitude reservoir. Compared with the reservoir at the same latitude in the low-altitude area with the same regulation performance and scale, the reservoir on the Qinghai-Tibetan Plateau had shorter stratification periods and weaker stratification intensities.

The seasonal cycle of stratification and mixing of the two reservoirs was ultimately determined by the seasonal changes in the net heat flux received by the surface layer. From September to December 2016, the average net heat flux of the Pangduo Reservoir and the low-altitude reservoir was -57.38 W/m$^2$ and -50.84 W/m$^2$ (Fig 5), respectively. The reservoirs began to lose heat, and inverse stratification intensified; then, the density difference between the epilimnion and metalimnion decreased. Once the surface temperature decreased, the epilimnion thickened quickly, and because the temperature of the cooled surface layers was the same as a part of metalimnion, this part of the thermocline became the epilimnion [33]. Before the winter of April 2017, the Pangduo Reservoir was frozen upstream of the dam and the heat exchange method of the surface water layer changed from "water-air heat exchange" to "ice-water heat exchange". The average heat loss of the surface water was slight and stable (the mean was

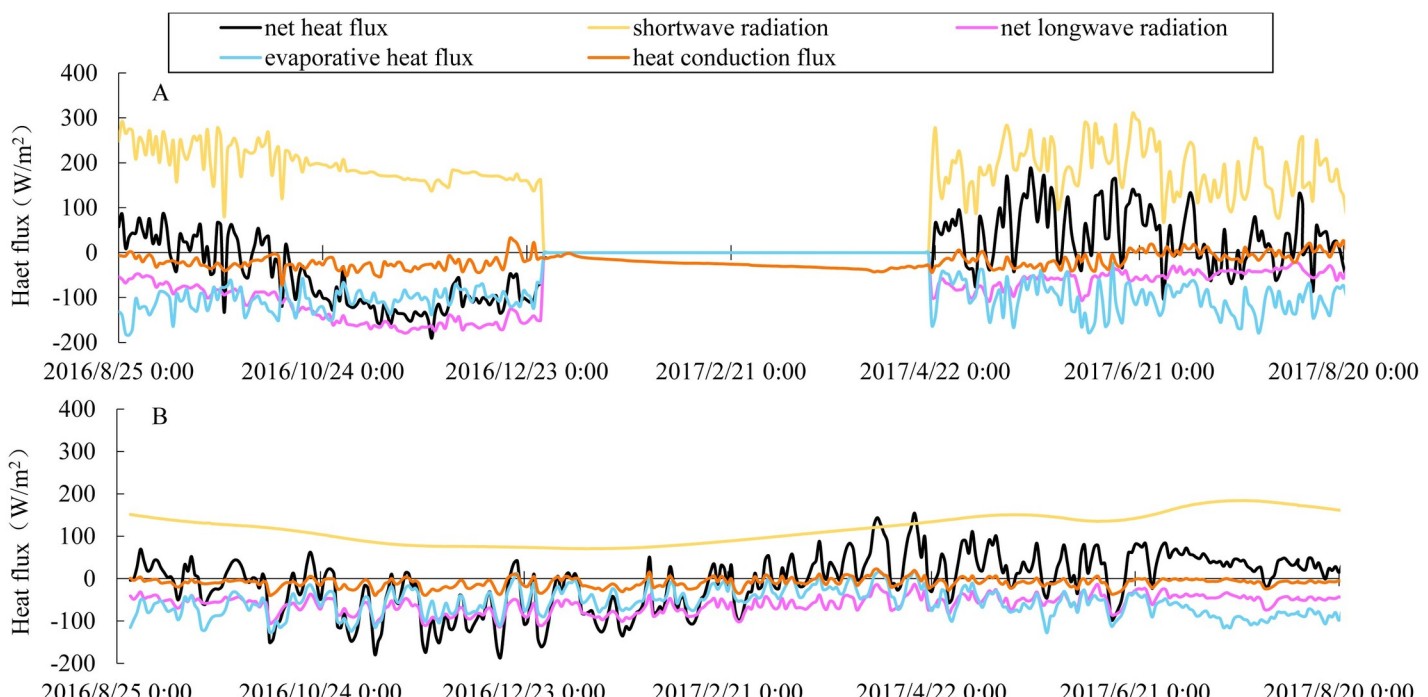

**Fig 5.** Comparison of the daily average heat budget of the surface water between the Pangduo Reservoir (A) and the low-altitude reservoir (B) during the calculated period.

-25.27 W/m$^2$). At the same time, the net heat flux was fluctuating and ranged from -160.80 W/m$^2$ to 154.37 W/m$^2$ in the low-altitude reservoir. From the end of April to August, the average net heat flux in the Pangduo Reservoir and the low-altitude reservoir were 37.11 W/m$^2$ and 35.12 W/m$^2$, respectively. The surface layer was endothermic, and the net heat flux was sufficient to make the difference in temperature and density between surface and deep layers [36].

Fig 6 shows the comparison of the diurnal heat budget of the surface water between the low-altitude reservoir and the Pangduo Reservoir on August 15, 2017. The epilimnion was affected by diurnal temperature fluctuations, heating during the daytime and cooling at night [58]. The net heat flux of the Pangduo Reservoir fluctuated drastically with a diurnal variation of 945.76 W/m$^2$, that of the low-altitude reservoir was 71.8% that of the Pangduo Reservoir. The diurnal variation was positive between 9 a.m. and 7 p.m. and reached the highest value at 1 p.m. in Pangduo Reservoir. Noticeable temperature increases and stratification was always observed in the epilimnion during sunlight hours [59]. Whether in the Pangduo Reservoir or the low-altitude reservoir, shortwave radiation had a greater impact on the net heat flux, which affected the trend and inflection point of the net heat flux. The shortwave radiation of the Pangduo Reservoir was strong, and the highest value of 898.34 W/m$^2$ was observed at 1 p.m., while the shortwave radiation of the low-altitude reservoir had a high value of 542.92 W/m$^2$ at 11 a.m. The net longwave radiation of the Pangduo Reservoir was generally smaller than that of the low-altitude reservoir, with an average daily difference of 35.5 W/m$^2$. The evaporation heat flux accounted for a large proportion of heat loss, which was higher in the Pangduo Reservoir than the low-altitude reservoir. The average value was 125.58 W/m$^2$ in the Pangduo Reservoir, which was 29.3% higher than that of the low-altitude reservoir. Heat conduction was slight in both the Pangduo Reservoir and low-altitude reservoir and caused differences in the amplitude and phase of the net heat flux.

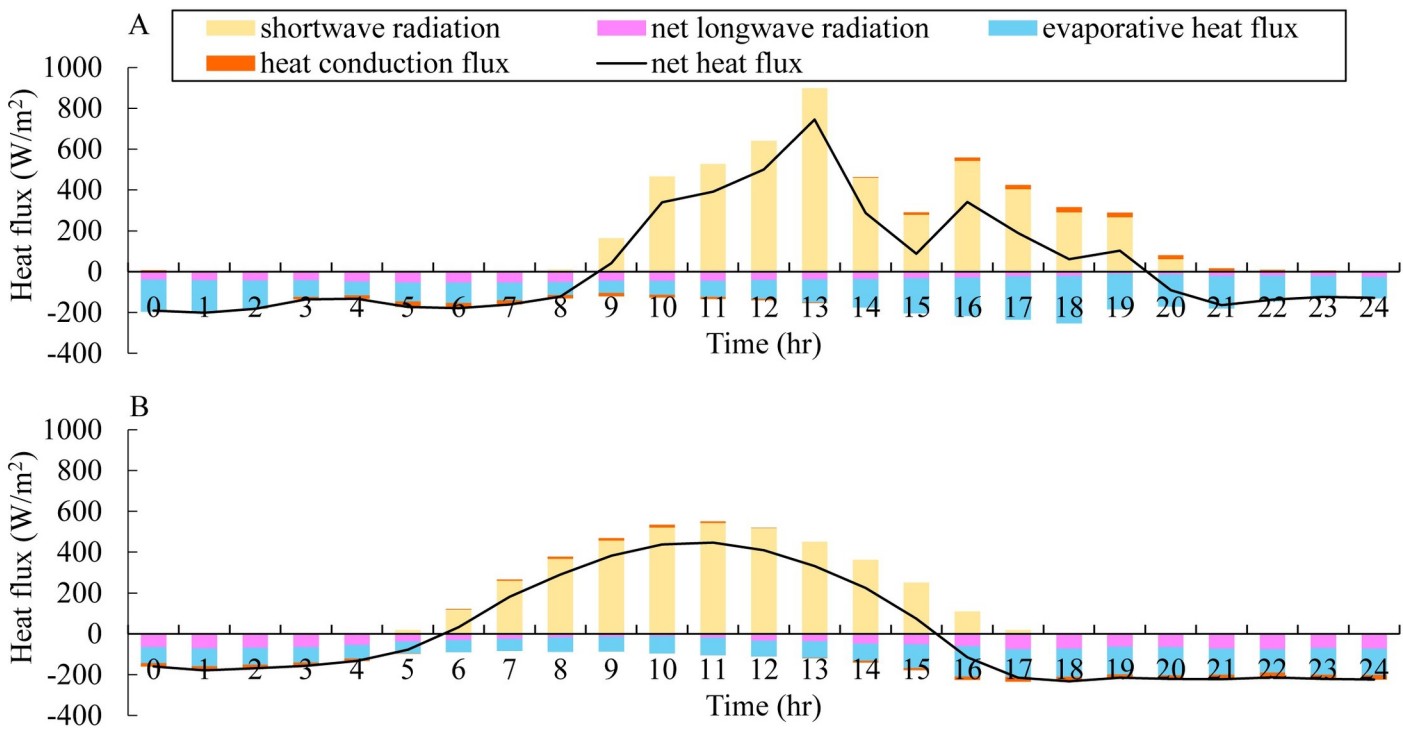

**Fig 6.** Comparison of diurnal heat budget of the surface water between the Pangduo Reservoir (A) and the low-altitude reservoir (B) on August 15, 2017.

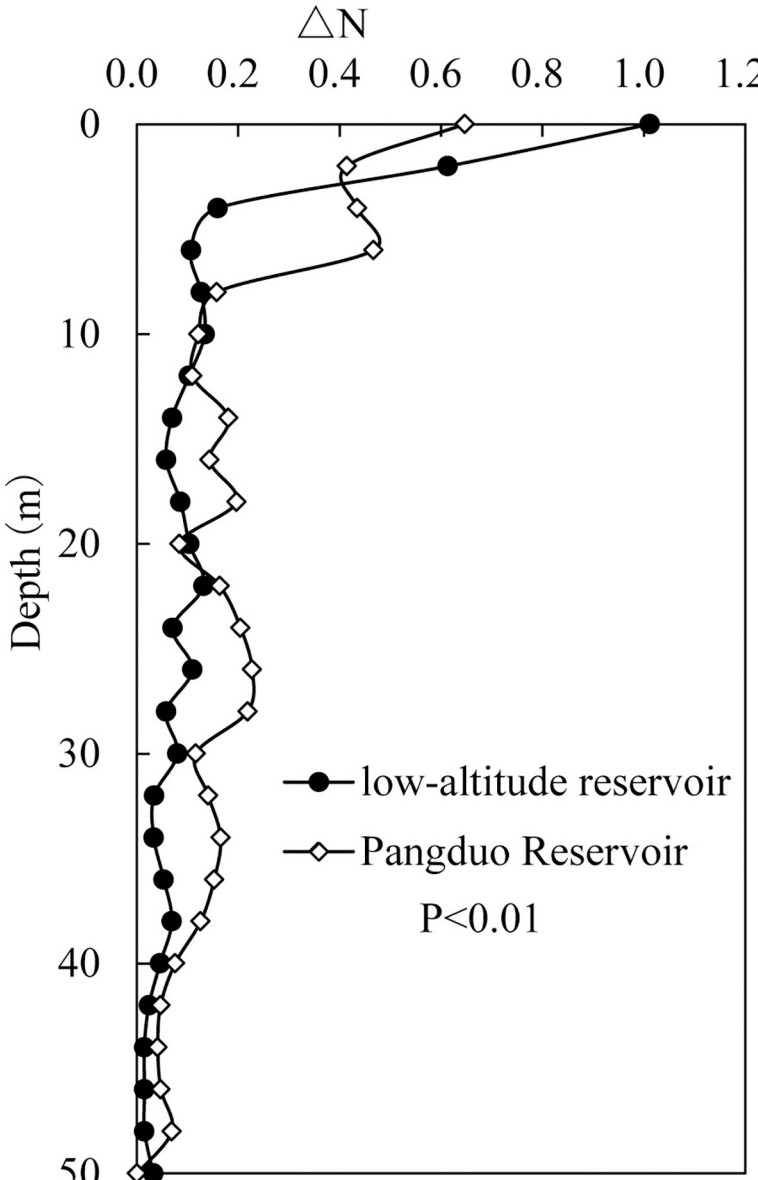

**Fig 7. Comparison of the daily variation in the buoyancy frequency of the water column in each layer upstream of the dam in the low-altitude reservoir and the Pangduo Reservoir.**

Fig 7 shows a comparison of the daily variation in the buoyancy frequency of the water column in each layer in front of the dam in the low-altitude reservoir and the Pangduo Reservoir during the high-temperature period on August 15, 2017. Significant differences were observed between the daily variation in the buoyancy frequency of the low-altitude reservoir and the Pangduo Reservoir ($P<0.01$). In the low-altitude reservoir, the diurnal variation stayed within only 5 m of the surface layer and the average diurnal buoyancy frequency difference was 0.59 $s^{-1}$. This finding was because the radiant heat loss from the surface layer at night was inadequate to transform sufficient potential energy into turbulent kinetic energy, such as in the Qinghai-Tibetan Plateau reservoir, thereby forming a turbulent penetrating convection effect. Similar to most low-altitude reservoirs, it still showed a stable buoyant flow regime [60, 61]. At

the same time, the inflow water temperature in the low-altitude reservoir was stable and varied between 28.0˚C and 28.3˚C (S1 File). The inflow steadily entered the mainstream layer along the subduction line, which was different from the special intraday inflow mixing model of the Qinghai-Tibetan Plateau.

### 4.6 Response of the thermal regime to future climate warming

Many previous studies have shown that lakes and reservoirs are sensitive to climate changes [62]. A stratified thermal structure is a basic physical feature of deep lake reservoirs that affects the material exchange inside the water body, such as that of nutrients and dissolved oxygen [19, 22, 63]. Climate warming would change the thermal regime and mixing of lakes and reservoirs, strengthen thermal stratification during the summer, increase thermal stability, and lengthen the stratification period [20, 64]. The strengthened thermal stratification in summer would cause a series of environmental impacts, especially those involving dissolved oxygen [65]. The lengthened period of thermal stratification would increase the duration of oxygen stratification and affect oxygen consumption of the hypolimnion [66]. For deep lakes and reservoirs with seasonal vertical mixing, the mixing process after cessation of stratification is an important mechanism for oxygen supplementation in the hypolimnion [67].

The Qinghai-Tibetan Plateau has a simple ecological structure and exhibits slow growth of aquatic organisms. Therefore, climate change will have a greater impact on the aquatic ecology [39–41]. Previous studies have shown that the stratification of lakes in the Qinghai-Tibetan Plateau increased by 6 days every 10 years under the conditions of climate warming [68]. However, man-made reservoirs formed by dams were similar to lakes and runoff, but their thermal regime and internal mixing mechanisms were different [69]. Further research is necessary to understand the impact of climate change on the thermal regime and mixing of man-made reservoirs on the Qinghai-Tibet Plateau. This research on the thermal regime of the Pangduo Reservoir has provided a relevant scientific basis and reference for the adaptive management of reservoirs on the Qinghai-Tibetan Plateau and similar high-altitude areas with regard to their response to climate change.

## 5. Conclusions

This study demonstrated that the reservoir of the Qinghai-Tibetan Plateau is a typical dimictic reservoir, and the water column had two turnovers: one at the end of March and one at the end of October. The strong diurnal variation in the meteorology and the inflow mixing mode strengthened the vertical convection mixing in the epilimnion during the stratified period; thus, a diurnal thermocline appeared. The mean variance of diurnal water temperature in the water column was 0.04, while that within 5 m was 0.10. In winter, the diurnal variation was weak due to the thermal protection of the ice cover. Compared with the low-altitude reservoirs at the same latitude, which had the same regulating performance and scale, the thermal regime exhibited distinct differences. The reservoir of the Qinghai-Tibetan Plateau had a shorter stratification period and weaker stratification intensity than the low-altitude reservoir. The annual average $SI$ value of the Pangduo Reservoir was 26.4 kg/m$^2$, which was only 7.5% that of the low-altitude reservoir. The seasonal changes in the net heat flux received by the surface layers ultimately determined the seasonal cycle of stratification and mixing of the two reservoirs. In this study, variations in the thermal regime and diurnal variation inside the reservoir on the Qinghai-Tibetan Plateau were investigated, which provided helpful information for adaptive management and decision-making for similar reservoirs in cold and high-altitude areas.

## Supporting information

**S1 Fig.** Temporal distribution of measured air temperature (A), solar radiation (B), and inflow water temperature (C Rezhenzangbu and D Wululong) in the study area.
(TIF)

**S2 Fig. Interpretation of the winter satellite film of the Pangduo Reservoir.**
(TIF)

**S3 Fig. Model grid division schematic diagram.**
(TIF)

**S4 Fig. Designed operating conditions of the Pangduo Hydropower station.**
(TIF)

**S5 Fig. Air temperature and inflow water temperature distribution of the low-altitude reservoir during the calculated period.**
(TIF)

**S6 Fig. Measured vertical water temperature changes from August 25, 2016 to August 31, 2016.**
(TIF)

**S7 Fig.** Simulated temperature profiles at the section upstream to the Pangduo dam using three shades (A), three WSCs (B) and measured profiles on Sep. 15, 2016; Oct. 15, 2016; Nov. 22, 2016 and Dec. 15, 2016.
(TIF)

**S8 Fig. Comparison of the calculated and measured water temperature in front of the dam (□ measured point).**
(TIF)

**S9 Fig. Measured and calculated water temperature scatter plot distribution.**
(TIF)

**S10 Fig. Variation process of the calculated ice thickness upstream of the dam.**
(TIF)

**S11 Fig. Longitudinal and vertical two-dimensional water temperature and flow field distribution in the Pangduo Reservoir on February 15, 2017.**
(TIF)

**S1 File. Inflow mixing mode videos and gifs between the Pangduo Reservoir and the low-altitude reservoir.**
(ZIP)

## Acknowledgments

We thank Minne Li, Jingying Lu, Hong Zhang, Wenyan He, and Jingting Wang for helpful discussions. We would also like to thank Ning Sun (academic editor), Hazel Bautista (handling editor) and three anonymous reviewers for providing comments on earlier versions of this manuscript.

## Author Contributions

**Conceptualization:** Yun Deng, Youcai Tuo.

**Data curation:** Yanjing Yang, Yun Deng, Youcai Tuo, Tianfu He.

**Funding acquisition:** Yun Deng, Jia Li.

**Investigation:** Yanjing Yang, Yun Deng, Youcai Tuo, Jia Li, Tianfu He.

**Methodology:** Yanjing Yang, Yun Deng, Jia Li, Tianfu He, Min Chen.

**Project administration:** Yun Deng.

**Software:** Yanjing Yang, Yun Deng, Youcai Tuo, Tianfu He, Min Chen.

**Supervision:** Yun Deng, Youcai Tuo, Jia Li.

**Validation:** Yanjing Yang, Yun Deng, Youcai Tuo, Jia Li.

**Writing – original draft:** Yanjing Yang.

**Writing – review & editing:** Yanjing Yang, Yun Deng, Youcai Tuo, Min Chen.

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
