## [Decision Letter · Decision Letter 0]

11 Aug 2020

PONE-D-20-19898

Study of the thermal regime of a reservoir on the Qinghai-Tibet Plateau, China

PLOS ONE

Dear Dr. Deng,

Thank you for submitting your manuscript to PLOS ONE. After careful consideration, we feel that it has merit but does not fully meet PLOS ONE’s publication criteria as it currently stands. Therefore, we invite you to submit a revised version of the manuscript that addresses the points raised during the review process.

We look forward to receiving your revised manuscript.

Kind regards,

NING Sun

Academic Editor

PLOS ONE

Journal Requirements:

Reviewers' comments:

Reviewer's Responses to Questions

**Comments to the Author**

1. Is the manuscript technically sound, and do the data support the conclusions?

Reviewer #1: Yes

Reviewer #2: Partly

Reviewer #3: Yes

2. Has the statistical analysis been performed appropriately and rigorously? 

Reviewer #1: Yes

Reviewer #2: No

Reviewer #3: No

3. Have the authors made all data underlying the findings in their manuscript fully available?

Reviewer #1: No

Reviewer #2: No

Reviewer #3: Yes

4. Is the manuscript presented in an intelligible fashion and written in standard English?

Reviewer #1: Yes

Reviewer #2: No

Reviewer #3: Yes

5. Review Comments to the Author

Reviewer #1: Study of the thermal regime of a reservoir on the Qinghai-Tibet Plateau, China

Review comments:

The CE-QUAL-W2 model is used to explore the typical thermal regime of a stratified reservoir on the Qinghai-Tibetan Plateau (Pangduo Reservoir). The study is interesting, and the modeling results are useful for the scientific understanding of the thermal dynamics in stratified reservoirs under the special geographical and meteorological conditions on the Qinghai-Tibetan Plateau. The paper is well organized, and the results are well presented and discussed. However, after screening the manuscript, I think there are still some points that can be improved. The detailed comments are as follows:

(1)In the Introduction, the CE-QUAL-W2 model should be mentioned in line 14 as it is a famous model.

(2)In lines 72-79, the authors should briefly list the reasons that why CE-QUAL-W2 model is used.

(3)I suggest the authors to summarize all the governing equations (1-8) to a table.

(4)Model grid deserves a figure.

(5)The authors should improve the conclusion section to make it more precise.

Based on the above comments, a minor revision is needed.

Reviewer #2: This study adapted CE-QUAL-w2 and simulated the 2-D sub-daily thermal structure of a high-altitude reservoir in Qinghai-Tibetan Plateau. The authors aim to further our understanding of a reservoir’s diurnal variation of a dimictic reservoir. Generally speaking, this draft is very raw and requires much work. I will reject it. However, the topic is interesting so I recommend resubmission.

Detailed comments are attached below.

Reviewer #3: The manuscript explains the thermal regime of a reservoir in Qinghai-Tibetan Plateau, China using the CE-QUAL-W2 model. The authors have used this 2D model in order to simulate the water temperature structure in the reservoir and verified their simulations against observed data. The authors found that the model could accurately simulate the water temperature and reported that the solar radiation was the major driver for the reservoir’s behavior with a shorter stratification period and weaker stratification stability. Some concerns first should be addressed. I have provided some broader comments here and some detailed comments in the attached document, all should be addressed properly.

General Comments:

1-Overall, I couldn’t find something new in the manuscript and based on the defined objective. If there is something that the authors think it was the key finding, they should clearly mention that in the text, in different parts of the manuscript.

2-The abstract doesn’t have even one sentence referring to the findings in a quantitative way. This is not acceptable. The authors should provide several interesting findings in a numerical way including different kinds of comparisons with percent change, R2, or p-value.

3-From the abstract, it seems the authors haven’t used a systematic statistical analysis. I would suggest they use a paired t-test and report the p-values.

4-The last paragraph of the introduction should include the main part of the manuscript representing the target, contribution, and science questions the manuscript is trying to answer. The current form of this paragraph is not informative enough and doesn’t have the required sections. The authors, for example, could address my first comment here. They also need to provide some explicit science questions in that paragraph and answer that in the results and discussion and highlight those findings in the abstract.

5-The authors have discussed the validation process and period in the methods however, there is no explanation about the calibration. What period and parameters have been used for the calibration and what was the stats out of the process against the validation? These all should be explained in the manuscript.

6-The results section includes several sub-sections at the beginning that are methods for me. This includes Fig. 2, Fig. 3, and Table 2. Please see my detailed comments in the attached document for more information.

7-The authors need to provide a subsection in the results to explain the sensitivity of the simulations in a systemic way. What was the most sensitive parameter for the simulations?

8-The manuscript (especially in the results section) includes too many figures and I believe some of them (For example Figs. 2 -5) could be moved to the supplementary material file.

6. PLOS authors have the option to publish the peer review history of their article (what does this mean?). If published, this will include your full peer review and any attached files.

Reviewer #1: No

Reviewer #2: No

Reviewer #3: No

---

## [Author Response · Author response to Decision Letter 0]

25 Sep 2020

Dear reviewer1,

We are grateful for the constructive comments and suggestions. Based on your comments, we revised the relevant content in the manuscript.

Our point-by-point responses to your comments are provided below.

1): In the Introduction, the CE-QUAL-W2 model should be mentioned in line 14 as it is a famous model.

Response to reviewer’s comment No. 1:

Thank you so much for your constructive suggestion. We added the CE-QUAL-W2 model in line 11 and reorganized the Abstract.

2): In lines 72-79, the authors should briefly list the reasons that why CE-QUAL-W2 model is used.

Response to reviewer’s comment No. 2:

Thank you so much for your suggestion. We listed the reasons for using CE-QUAL-W2 in lines 83-89 and added relevant references in the Introduction.

3): I suggest the authors to summarize all the governing equations (1-8) to a table.

Response to reviewer’s comment No. 3:

Thank you for your suggestion. We summarized all the governing equations in Table 2 in line 163. 

4): Model grid deserves a figure.

Response to reviewer’s comment No. 4:

Thank you so much for your comment. We added a figure to the model grid in the Supporting information (S3 Fig).

5): The authors should improve the conclusion section to make it more precise.

Response to reviewer’s comment No. 4:

Thank you so much for your constructive comment. We reorganized the Conclusions in lines 407-420 according to your suggestion.

Dear reviewer2,

We are grateful for the constructive comments and suggestions. Based on your comments, we revised the relevant content in the manuscript. We reorganized the manuscript as you suggested.

(1) We added section numbers.

(2) We reorganized the Introduction according to your comments.

(3) We explained the reasons for selecting the three metrics (VTG, N, and SI) and added relevant references.

(4) We sent our manuscript to a professional English editing service (American Journal Experts (AJE), USA) to improve the English throughout the manuscript. AJE has made changes to reflect standard conventions for phrasing, grammar, and punctuation, and we ensured that the intended meaning has been maintained. We modified our manuscript according to your constructive comments, conducted relevant statistical significance tests and reported the P values. Moreover, we added a sensitivity analysis section of the model.

(5) We reorganized and interpreted the results according to your comments. 

Our point-by-point responses to your comments are provided below.

Abstract: 

1): Line 9: “…meteorological characteristics with low temperature…” it is better to clarify ‘low air temperature’ because it can be confused with reservoir water temperature.

Response to reviewer’s comment No. 1:

Thank you for this suggestion. We modified this text in line 9.

2): Line 13: Please simply explain what a “dimictic reservoir” is. This term is not commonly acknowledged.

Response to reviewer’s comment No. 2:

Thank you for your constructive comment. The dimictic reservoir is covered with ice during part of the year and stratified in the other part, with two mixed periods in between. We specified the mixed period in line 17 and explained the “dimictic reservoir” in lines 66-67 in the Introduction.

3): Please provide a summary of your model performance. Do not use the statement, i.e., “the model accurately simulated…”. Readers can determine whether your model is accurate or not based on your model performance summary.

Response to reviewer’s comment No. 3:

Thank you for this suggestion. We provided the quantitative results of the model validation in lines 13-14 in the Abstract. 

Introduction:

4): “Moreover, early research focused primarily on regional differences.” I don’t think this is correct.

Response to reviewer’s comment No. 4:

Thank you so much for your suggestion. We meant to express the difference between the diurnal variations of tropical and temperate reservoirs. We rewrote this part in lines 61-68 and added relevant references.

5): Line 38: “however, the lack of detailed data samples and perfect research methods reflect certain existing limitations” This statement is ambiguous, and no research methods are perfect.

Response to reviewer’s comment No. 5:

Thank you for your comment. We agree with your idea. We rewrote this part and supplemented the model methods used in related research and relevant references in lines 83-89.

6): Line 62-64: Wordy. I would recommend changing to “the Qinghai-Tibetan Plateau is ecologically vulnerable to human activities and environmental changes.”

Response to reviewer’s comment No. 6:

Thank you for your recommendation. We replaced the text in lines 74-75.

7): Line 69: “few scientists seem to be aware”. It is not proper to use “seem to be” in a scientific paper.

Response to reviewer’s comment No. 7:

Thank you so much for your reminder. We modified this part according to your suggestions in lines 81-82.

8): Some sentences are very long and hard to follow, e.g., Line 47-49, Line 72-75, and Line 76-79.

Response to reviewer’s comment No. 8:

We rewrote these sentences in lines 53-55, lines90-97.

9): Line 46: "variable temperature layer" suggesting as "thermocline".

Response to reviewer’s comment No. 9:

We replaced this (line 48).

Study area:

10): Line 85: What is a “controlling project”?.

Response to reviewer’s comment No. 10:

“Controlling project” here refers to controlled water conservancy projects, which are projects constructed for the purpose of controlling and arranging surface water and groundwater in the natural world to eliminate harm and provide benefits. We rewrote this part in lines 102-104 for clarification.

11): Line 86: What is a normal storage level? Is it necessary in this study?

Response to reviewer’s comment No. 11:

Normal water storage levels refer to the highest water level stored by the reservoir under safe operation conditions. We deleted this phrase, and the engineering characteristics of the reservoir have been reorganized in lines 104-111.

12): Line 87-88: Unit for “total water area” is wrong. Additionally, the water area and depth in front of dam should vary with the amount of water stored in reservoirs.

Response to reviewer’s comment No. 12:

Thank you for your reminder. We corrected the unit error in line 111. The water area and depth do vary with the water stored in reservoirs. We rewrote this part in lines 108-111.

13): Line 97: Please switch Rezhenzangbu and Wululong to be consistent with Figure numbering..

Response to reviewer’s comment No. 13:

We switched them in line 120.

14): Table 1: B1: Please use “Upstream of dam” instead of “In front of dam”.

Response to reviewer’s comment No. 14:

Thank you for your suggestion. We replaced this text in Table 1 and other relevant places.

Mathematical models and methods:

15): Line 119-121: Try not to use passive voice.

Response to reviewer’s comment No. 15:

Thank you for your suggestion. We rewrote this text in lines 152-153.

16): Line 123: What variables are you referring to?.

Response to reviewer’s comment No. 16:

The variables specifically refer to the flow velocity and the dispersion coefficient along the width of the river, which have been ignored in the equations.

17): Line 133: “…, Bη the water surface width” where η should be subscript

Response to reviewer’s comment No. 17:

Thank you for your reminder. We corrected this text (Table 2).

18): Line 134: is �� and �GY "# the same thing? If so, can you make the notation consistent in this paper?

Response to reviewer’s comment No. 18:

Thank you for your comments. They are not the same parameter: �� is the heat transfer source term between water bodies and �GY is the heat exchange term between water and air.

19): Line139: please add reference for this equation.

Response to reviewer’s comment No. 19:

Reference 46 is the source of these governing equations. We added this text in line 160.

20): Line 144: Following terms i.e., the longitudinal eddy viscosity coefficient and the longitudinal eddy current diffusion coefficient, do not occur anywhere in previous context. Are you referring to longitudinal and vertical dispersion coefficients as in Line 134? If so, please be consistent with terminology throughout the paper. Additionally, please explain why you chose those two values.

Response to reviewer’s comment No. 20:

Thank you for your comments. This part has been deleted. We added content on the parameter calibration in lines 210-217.

21): Line 146: What is formula W2N?

Response to reviewer’s comment No. 21:

Thank you for your comments. This part has been deleted. The W2N formula is a mixed length formula, which is one of the calculation formulas for the vertical vortex viscosity coefficient. It does not belong to the governing equation and has been deleted in the manuscript.

22): Line 164: “Although N is the convention used in limnology and oceanography, reservoirs are similar.” The buoyancy frequency has been used in manmade reservoirs for a very long time, e.g., Snodgrass& O’Melia (1975), Niemeyer et al., (2018).

Response to reviewer’s comment No. 22:

Thank you for your comments and recommended references. We rewrote this sentence in line 180.

23): 11: I think it is more generalizable to use an integral instead of summing.

Response to reviewer’s comment No. 23:

Thank you for your suggestion. We wrote this as an integral instead in equation 10.

24): Line 177-179: I don’t understand.

Response to reviewer’s comment No. 24:

Water age (days) depicts the duration of time that water has stayed in a waterbody, and it is defined as the persistence of water after it enters a reservoir from upstream. In the Pangduo CE-QUAL-W2 model, we set a virtual constituent as a state variable that has an initial value of 0 when entering the reservoir and decays by 1 per day and does not interact with any other water quality state variables. This variable represents the water age. By using the water age, we can identify the transport of the inflow water and the corresponding mixing processes. The water age of new water flowing into the reservoir is always less than the age of the ambient water already in the reservoir. For instance, in the case of overflow in a reservoir, the fresher water in the upper layer has a lower water age than the ambient water in the lower layers.

Results and discussion:

25): Line 182: Mean annual air temperature? Be specific

Response to reviewer’s comment No. 25:

Thank you for your suggestion. We reorganized this part in lines 129-130 according to your comments.

26): Fig. 6 can be as supplemental information.

Response to reviewer’s comment No. 26:

Thank you for your suggestion. Fig 6 has been renamed Fig S4 of the Supplementary material. The manuscript has also been modified accordingly (line 172).

27): Line 186-189: You mentioned solar radiation twice with description of longwave radiation in between. Please do not go back and forth and explain solar radiation first and then longwave radiation. Also why do you only explain the diurnal variation of solar radiation only? Why not talk about the diurnal variation of longwave radiation?

Response to reviewer’s comment No. 27:

Thank you for your suggestion. We rewrote this part in lines 132-135. Here, we mainly explain the monitoring data of our measured site. Longwave radiation was not part of our monitoring project; thus, it was not explained here.

28): Line 193: Define “daily variation”. How do you calculate that? This is an important variable and used many times in this study. A definition is necessary.

Response to reviewer’s comment No. 28:

Thank you so much for your suggestion. Diurnal variation has been defined in the Introduction in lines 56-58, and it specifically refers to the epilimnion heated during the daytime and cooled at night and is subject to the influence of temperature fluctuation every day and night. We used indexes (VTG, N) and statistical methods for the evaluation.

29): Line 198: Be careful with the word “significant”. This word usually implies that the authors have conducted a statistical significance test and the trend/result they find is statistically “significant”. “Strong diurnal variation” is better.

Response to reviewer’s comment No. 29:

Thank you for this reminder. We replaced this term with “strong” in line 201 and modified other similar situations in the manuscript.

30): Line 210-212 As shown in Table 1, measured vertical water temperature was available from 08/25/2016 to 08/31/2016. How did you get measured temperature data for 09/15, 10/15, 11/22, and 12/15? Additionally, as shown in Figure 3, the temperature difference between depth at 0 m and depth at 30 m can be as large as 8 degree Celsius. It is incorrect to say, “temperature stratification of the reservoir is weak,” since reservoirs should be more stratified at warmer seasons. Based on Section Monitoring data analysis, hottest month is July, so it would make the model more convincing if we can see the model performance for a hot July day. I am not sure whether that data is available or not, but an alternative way is to show the model performance for 08/25 to 08/31 when thermal stratification is stronger than 09/15.

Response to reviewer’s comment No. 30:

Thank you so much for your suggestion. The temperature data for 09/15, 10/15, 11/22, and 12/15 were measured by EXO2 (Table 1) during manual monitoring times, and we explained the data source in lines 218-219. We agree that “temperature stratification of the reservoir is weak” is inappropriate here, and we modified it in lines 219-220. Unfortunately, we do not have the profile monitoring data for July. In the model verification, we added the vertical water temperature comparison between August 25 and August 30 (S7 Fig) and re-analyzed the error in lines 222-224.

31): Line 215-216: The metric shown in the paper is also misleading since the errors can be diluted by the winter days when thermal stratification is weak. For example, we can see that, on 09/15/2016, model simulated a stronger stratification than the measurement (Figure 4).

Response to reviewer’s comment No. 31:

Thank you for your comments. In fact, excluding the winter data, the absolute error was 0.27℃, relative error was 2.2%, standard deviation was 0.369 and root mean square error was 0.478℃; thus, the data can illustrate the accuracy of the simulation. The situation you mentioned does exist because the model itself cannot perfectly reflect the actual process, although the temperature distribution and change process are acceptable. We also redrew Fig 4.

32): Figure 5: I cannot tell the model performance from this plot. This only shows the model simulation result. Please show measurement in this plot.

Response to reviewer’s comment No. 32:

Thank you so much for your comments. Because of engineering limitations and researcher safety considerations, we do not have measured data for the winter icing period. Therefore, we obtained the ice process in the study area through satellite pictures, and ice was also considered in the model. Judging from the time of freezing and melting, the values were consistent, which shows that the process of thermal budget was consistent. We rewrote this part in lines 229-232 and moved Fig 4 to the Supporting information (S9 Fig).

33): Line 240: Be consistent with the unit of Buoyance Frequency (N). You defined the unit of N as 1/s in Line 159.

Response to reviewer’s comment No. 33:

Thank you so much for your reminders. We replaced it in line 267 and checked and revised other similar text in the manuscript.

34): Line 249-250: Expression is unprofessional. Please rephrase.

Response to reviewer’s comment No. 34:

Thank you so much for your suggestion. We rephrased the expression in lines 253-256.

35): Line 269: Sooner than what?

Response to reviewer’s comment No. 35:

Thank you so much for your suggestion. We rewrote this sentence in lines 272-274.

36): Line 269-270: The retention time of reservoirs is irrelevant to elevations. Retention time is a measure of how long water resides in reservoirs, roughly depending on reservoir storage and outflow. Reservoirs at low altitude can have either very short or very long residence times. You may refer to Cheng et al. (2020), Yearsley et al. (2019), Yigzaw et al. (2019).

Response to reviewer’s comment No. 36:

Thank you for your comments and recommendations. We agree with you that the retention time of reservoirs is irrelevant to the elevation but related to the thermal regime. We meant to express the difference between warm monomictic reservoirs and dimictic reservoirs; thus, we rewrote this part in lines 270-274.

37): Line 280: If the authors did not conduct statistical significance test, please use “strong” or other words instead of “significant”.

Response to reviewer’s comment No. 37:

Thank you for your suggestion. We replaced the text and performed a variance analysis in lines 281-282.

38): Line 285: If the time step is hourly, the minutes are not necessary, i.e., 8AM will do. Additionally, please use 12AM instead of 0AM.

Response to reviewer’s comment No. 38:

Thank you for your reminder. We modified the text and checked the other text in the manuscript and relevant figures.

39): Second paragraph in Section Diurnal characteristics analysis is poorly organized and hard to follow. The authors jump back-and-forth between top 5-m range and top 20-m range, so the values are hard to be compared across different time slots.

Response to reviewer’s comment No. 39:

Thank you for your suggestion. We reorganized this analysis according to your comments in lines 282-295. 

40): Figure 7 is boring and not informative.

Response to reviewer’s comment No. 40:

Thank you for your suggestion. We carefully considered your suggestions, re-produced the GIF and videos to support the analysis here (S1 File), and re-analyzed the inflow mode in lines 303-309.

41): Line 310-311: Recommend not using one sentence as a paragraph Figure 9 is boring. The authors can simply say that diurnal variation is weak in winter...

Response to reviewer’s comment No. 41:

Thank you for your reminder. We corrected the text and re-analyzed this part in lines 310-315 according to your comments. Fig 9 has been deleted in the manuscript and moved to the Supporting material (S10 Fig).

42): Figure 10: Why is there no shortwave radiation from 12/23/2016 to 04/22/2017? Figure S1 shows that at least one-third of reservoir surface was not covered with ice even at coldest time (02/18/2017). In CE-QUAL-w2 model, did you assume that all reservoir surfaces are uniformly covered with ice?

Response to reviewer’s comment No. 42:

Thank you for your comments. The result of the heat budget here is the section upstream of the dam (Fig1 B1). This section was frozen during the monitoring period; therefore, shortwave radiation was not considered during this period. The model does not assume that all sections are covered by ice. We modified this part in lines 356-369.

43): Line 356: How to interpret SI value? Does bigger value mean less stability?

Response to reviewer’s comment No. 42:

Thank you so much for your comments. A bigger SI indicates that greater energy is required to achieve vertical mixing of the water column and represents the more stable stratification. We provided an explanation and performed relevant statistical analyses in lines 343-351.

Dear reviewer3,

We greatly appreciate the constructive comments and suggestions. We have reorganized the manuscript as you suggested.

(1) We rewrote the Abstract in a quantitative way according to your comments, defined the objective and highlighted our findings.

(2) We performed a paired t test to analyze the relevant results and reported the P values.

(3) We carefully considered your suggestions for the Introduction and modified it with the required sections as you mentioned.

(4) We supplemented the calibration process in the manuscript and the relevant figure and explained the sensitivity of the simulations.

(5) We reorganized the structure of the manuscript and moved some figures (Figs 2-5) to the supplementary material file according to your comments. 

Our point-by-point responses to your comments are below.

1): Lines 14-16: Provide numerical results here.

Response to reviewer’s comment No. 1:

Thank you for this suggestion. We modified it with quantitative results in line 13-14.

2): Lines 16-17: What was the dT? What was the percent change compared to winter time?

Response to reviewer’s comment No. 2:

Thank you for your comment. We added the results of the analysis of variance here for illustration purposes in lines 18-19.

3): Line 18: What was the number?

Response to reviewer’s comment No. 3:

We added the results of the analysis of variance in lines 20-22.

4): Lines 19-20: Provide the percentages.

Response to reviewer’s comment No. 4:

Thank you for your suggestion. We added percentages and the P values of the paired t test in lines 22-25.

5): Lines 20-21: Again support this argument with quantitative results.

Response to reviewer’s comment No. 5:

Thank you for your comment. We rewrote this part in lines 25-26.

6): Line 8: Please see my comments in the 1st page.

Response to reviewer’s comment No. 6:

Thank you for your constructive recommend. We reorganized it according to your comments. We clarified the research objective and key findings and referenced the results in a quantitative way (lines 9-29).

7): Line 28: Checking your references in this paragraph, they are mostly old and out of date. Use some newly published papers in the field, here are some examples:

https://doi.org/10.1016/j.jenvman.2019.110023

https://doi.org/10.1016/j.scitotenv.2019.03.248

https://doi.org/10.3390/w11051060.

Response to reviewer’s comment No. 7:

Thank you so much for your reminder and recommendations. We added these recommended references and replaced some outdated references in the Introduction (references 2-6, 8, 11, 45, 47-49).

8): Line 75: Mention the name of the model in the abstract as well

Response to reviewer’s comment No. 8:

We mentioned the name in the Abstract in line 11.

9): Line 92: You should provide a more informative caption and clearly mention the location of the reservoirs separately in the map and in the caption.

Response to reviewer’s comment No. 9:

Thank you for your suggestions. We rewrote this part with more specific information, defined the location in lines 102-113, and re-drew Fig 1.

10): Line 117: This in the 3rd time I am seeing this in the text. You should avoid redundancy in the text. Consider this and similar issues throughput the manuscript.

Response to reviewer’s comment No. 10:

Thank you so much for your suggestions. We have reorganized this paragraph in lines 151-158 and checked and revised similar issues in the manuscript carefully. We have also sent our manuscript to a professional English editing service (American Journal Experts (AJE), USA) to improve the English throughout the manuscript.

11): Line 120: Provide a proper citation for the model.

Response to reviewer’s comment No. 11:

We have added the reference (Reference 46) and revised the text in line 153.

12): Line 126: If you haven't updated any of the following equations, you don't have to provide all of them as they are already published and accessible. You can remove some or all of them or send them to a supplementary material file to save some space.

Response to reviewer’s comment No. 12:

Thank you so much for your suggestions. The governing equations were still retained for easy of reference, but we have summarized the governing equations (1-8) to Table 2 to save space.

13): Line 150: How about calibration?

Response to reviewer’s comment No. 13:

We have supplemented the calibration process in lines 210-217 in the manuscript and the relevant figure (S6 Fig), and explained the sensitivity coefficient of the simulations.

14): Line 161: Provide citation for this and Eq. 10.

Response to reviewer’s comment No. 14:

Thank you for your reminder. We have added it in lines 175-176 with reference 50 and 51.

15): Line 182: Needs a space. Check all the document for similar issues.

Response to reviewer’s comment No. 15:

Thank you for your reminder. We have added it in line 129 and checked and revised similar issues in the manuscript.

16): Line 183: Add degree C.

Response to reviewer’s comment No. 16:

We have added it in line 129 and checked and revised similar issues in the manuscript.

17): Lines 182-194: This is not part of your results. It should be presented in the Methods section and the figures also could be presented in the supplementary materials file.

Response to reviewer’s comment No. 17:

Thank you for your reminder. We have moved this information to the section Regional monitoring in lines 129-138. The figures are now presented in the Supporting information (S1 Fig).

18): Line 196: Explain the panels in the caption.

Response to reviewer’s comment No. 18:

Thank you for your reminder. We have explained the panels in the captions in line 596.

19): Lines 216-217: Provide the numerical results here as well.

Response to reviewer’s comment No. 19:

Thank you for your reminder. We have provided the numerical results in lines 225-226.

20): Lines 210-211: Figure 4 only includes 4 panels without further information on why these 4 days are selected in the period. How was the performance of the model in calibration compared to the validation? If these values in fig. 4 are the average daily or observed for a time. Overall this figure is confusing and should be deleted. You need to think of a better way of visualizing this.

Response to reviewer’s comment No. 20:

Thank you for your comments. We have explained why these 4 days are selected in lines 218-220. The validation period included the high temperature period in summer and the low temperature period before freezing. The calibrated data were measured by EXO2 (Table 1) during the manual monitoring time, and we explained the data source in lines 218-219. In the model verification, we added the vertical water temperature comparison between August 25 and August 30. In addition, we redrew Figure 4 to clarify the visual presentation (S7 Fig) and re-analyzed the errors in lines 222-224.

21): Lines 222-229: This is a part of the methods for me.

Response to reviewer’s comment No. 21:

Thank you for your comments. This part was actually a verification of the ice period. We did not measure data during the winter icing period because of engineering limitations and researcher safety considerations. Therefore, we obtained the ice process in the study area through satellite pictures, and ice was also considered in the model. The time of freezing and melting was consistent, which showed that the process of thermal budget was consistent. We rewrote this part in lines 229-232 and moved Figure 4 to the supporting information (S9 Fig).

22): Lines 233-234: Present this in a quantitative way.

Response to reviewer’s comment No. 22:

Thank you for your comments. We rewrote this sentence in lines 236-239 using quantitative results.

23): Lines 279-280: By what percent?

Response to reviewer’s comment No. 23:

Thank you for your comments. We rewrote this information based on the variance analysis results in lines 281-282 and reorganized this paragraph.

24): Line 327: Provide relevant information for Fig. 10 first.

Response to reviewer’s comment No. 24:

We carefully checked and revised the order of the pictures to ensure that the order of the pictures and text matches.

25): Line 331: The panels are too small and again, there is not explanation why and how these time windows are selected.

Response to reviewer’s comment No. 25:

Thank you so much for your suggestion. We redrew Figure 4 and enlarged the panel according to your comments. We explained why and how these time windows are selected in lines 327-328 and used the recommended paired t-test to analyze this part of the data and reported the P value, and the results were reorganized and discussed in lines 343-351.

---

## [Decision Letter · Decision Letter 1]

26 Oct 2020

PONE-D-20-19898R1

Study of the thermal regime of a reservoir on the Qinghai-Tibet Plateau, China

PLOS ONE

Dear Dr. Deng,

Thank you for submitting your manuscript to PLOS ONE. After careful consideration, we feel that it has merit but does not fully meet PLOS ONE’s publication criteria as it currently stands. Therefore, we invite you to submit a revised version of the manuscript that addresses the points raised during the review process.

All three reviewers agreed that the revisions made significant improvements to the quality of the paper. One reviewer had minor comments that the authors should address in their revisions. 

We look forward to receiving your revised manuscript.

Kind regards,

NING Sun

Academic Editor

PLOS ONE

Reviewers' comments:

Reviewer's Responses to Questions

**Comments to the Author**

1. If the authors have adequately addressed your comments raised in a previous round of review and you feel that this manuscript is now acceptable for publication, you may indicate that here to bypass the “Comments to the Author” section, enter your conflict of interest statement in the “Confidential to Editor” section, and submit your "Accept" recommendation.

Reviewer #1: All comments have been addressed

Reviewer #2: All comments have been addressed

Reviewer #3: All comments have been addressed

2. Is the manuscript technically sound, and do the data support the conclusions?

Reviewer #1: Yes

Reviewer #2: Yes

Reviewer #3: Yes

3. Has the statistical analysis been performed appropriately and rigorously? 

Reviewer #1: Yes

Reviewer #2: Yes

Reviewer #3: Yes

4. Have the authors made all data underlying the findings in their manuscript fully available?

Reviewer #1: No

Reviewer #2: No

Reviewer #3: Yes

5. Is the manuscript presented in an intelligible fashion and written in standard English?

Reviewer #1: Yes

Reviewer #2: Yes

Reviewer #3: Yes

6. Review Comments to the Author

Reviewer #1: The authors have addressed all the review comments, and now the paper is acceptable for publication.

Congratuation for the good work.

Reviewer #2: General comments:

I would like to thank the authors made a tremendous effort in updating this manuscript. It has improved a lot. I would still recommend the authors address the following questions before it goes to publication.

Specific comments:

1. Line 14: please specify what frequency of the observed data is used to calculate RMSE? Hourly, daily, or monthly?

2. Line 90: What do you mean by the “river-run reservoir”? Do you mean “run-of-river reservoir”? Based on the data you showed in Figure 2d, I would argue whether this reservoir is a run-of-the-river reservoir. It is more like a storage reservoir. Commonly, run-of-the-river reservoirs are barely stratified.

3. Line 111: “…reservoir is 12.3 x 108 m3 [at] (under) the 4095 m a.s.l.”

4. Equation 11 is not necessary. The authors can simply state that in text.

5. Section 3.4: I would recommend the authors have a summary paragraph to summarize what statistical experiments they did in this study. The current paragraph is oversimplified, and I am not particularly interested in what software the authors used to do this analysis; Python, R, Excel does not make any difference.

6. Line 223: Add units to standard deviation. Additionally, it is unnecessary to report a relative error in river temperatures. Unlike precipitation, zero precipitation means there is no precipitation, but zero degrees Celsius still has physical meanings.

7. Please rearrange the supplemental figures as they appear in the main text.

8. Line 236: I would recommend using the term “turnover” rather than “inversion.”

9. Line 280: “…fluctuated [between] (from) XX °C [and] (to) XX°C.” The temperature fluctuation is not monotonic.

10. The first paragraph in Section 4.5 belongs to methodology.

11. It would recommend that the authors further discuss how this study's findings can have a broader impact. It would make this study more impactful.

Notation: [add] (delete)

Reviewer #3: The manuscript has been thoroughly revised and basically the manuscript has been improved. It may be published now.

7. PLOS authors have the option to publish the peer review history of their article (what does this mean?). If published, this will include your full peer review and any attached files.

Reviewer #1: No

Reviewer #2: No

Reviewer #3: No

---

## [Author Response · Author response to Decision Letter 1]

4 Nov 2020

Dear reviewers,

We are grateful to the editor for giving us the opportunity to further improve manuscript PONE-D-20-19898R1, entitled “Study of the thermal regime of a reservoir on the Qinghai-Tibetan Plateau, China”. We are grateful for the constructive comments and suggestions, which were carefully considered in this minor revision of our manuscript.

Our point-by-point responses to your comments are provided below.

1): Line 14: please specify what frequency of the observed data is used to calculate RMSE? Hourly, daily, or monthly?

Response to reviewer comment No. 1:

Thank you for this comment. We specified the frequency of the observed data in lines 13-14.

2): Line 90: What do you mean by the “river-run reservoir”? Do you mean “run-of-river reservoir”? Based on the data you showed in Figure 2d, I would argue whether this reservoir is a run-of-the-river reservoir. It is more like a storage reservoir. Commonly, run-of-the-river reservoirs are barely stratified.

Response to reviewer comment No. 2:

Thank you for your constructive comment. We meant “run-of-river reservoir”, and this was modified in line 91. Most of the inflow of run-of-river reservoirs enters from the main river and forms a plug flow with a mainstream direction. These reservoirs have a larger flow, a smaller volume and a mainstream layer, and the retention time is smaller than that of storage reservoirs and lakes. The Pangduo Reservoir has these characteristics. With the expansion of reservoir construction, these large run-of-river deep reservoirs will also have temperature stratification, such as that in the Xiluodu Reservoir, and the water age of the hypolimnion varies between 100 and 300 days. The specific references are listed below.

References:

Reference 1. Xie, Q., Liu, Z., Fang, X., Chen, Y., Li, C., MacIntyre, S., 2017. Understanding the Temperature Variations and Thermal Structure of a Subtropical Deep River-Run Reservoir before and after Impoundment. Water-Sui. 9(8), 603.

Reference 2. Hayes, N.M., Deemer, B.R., Corman, J.R., Razavi, N.R., Strock, K.E., 2017. Key differences between lakes and reservoirs modify climate signals: A case for a new conceptual model. Limnology & Oceanography Letters 2(2).

Reference 3. Naderi, V., Farsadizadeh, D., Dalir, A.H., Arvanaghi, H., 2014. Effect of Using Vertical Plates on Vertical Intake on Discharge Coefficient. Arabian Journal for Science & Engineering 39(12), 8627-8633.

Reference 4. Akiyama, J., Stefan, H.G., 1987. Gravity Currents in Lakes, Reservoirs and Coastal Regions: Two-Layer Stratified Flow Analysis. St Anthony Falls Laboratory.

3): Line 111: “…reservoir is 12.3 x 108 m3 [at] (under) the 4095 m a.s.l.”

Response to reviewer comment No. 3:

Thank you for your reminder. This was modified in line112 according to your comment.

4): Equation 11 is not necessary. The authors can simply state that in text.

Response to reviewer comment No. 4:

Thank you very much for your suggestion. We deleted Equation 11 and stated the content in the text in lines 199-202.

5): Section 3.4: I would recommend the authors have a summary paragraph to summarize what statistical experiments they did in this study. The current paragraph is oversimplified, and I am not particularly interested in what software the authors used to do this analysis; Python, R, Excel does not make any difference.

Response to reviewer comment No. 5:

Thank you for your constructive comment. We revised the paragraph to summarize the statistical experiments performed in the study in lines 204-209.

6): Line 223: Add units to standard deviation. Additionally, it is unnecessary to report a relative error in river temperatures. Unlike precipitation, zero precipitation means there is no precipitation, but zero degrees Celsius still has physical meanings.

Response to reviewer comment No. 6:

Thank you for your reminder. We added units to the standard deviation in line 234 and performed similar edits throughout the manuscript. We agree with you that reporting the relative error is unnecessary, and this was deleted.

7): Please rearrange the supplemental figures as they appear in the main text.

Response to reviewer’s comment No. 7:

Thank you for your reminder. We rearranged and carefully checked the order of supplementary figures in the manuscript.

8): 8. Line 236: I would recommend using the term “turnover” rather than “inversion”.

Response to reviewer comment No. 8:

We modified this term in line 246, line 248 and line 430.

9): Line 280: “…fluctuated [between] (from) XX °C [and] (to) XX°C.” The temperature fluctuation is not monotonic.

Response to reviewer comment No. 9:

Thank you very much for your reminder. This was modified in line 290 and line 309.

10): The first paragraph in Section 4.5 belongs to methodology.

Response to reviewer comment No. 10:

Thank you for your comment. We moved the first paragraph in section 4.5 to the methodology section, specifically, the last paragraph of section 3.2, lines 174-178.

11): It would recommend that the authors further discuss how this study's findings can have a broader impact. It would make this study more impactful.

Response to reviewer comment No. 11:

Thank you for your constructive comment. We added section 4.6 and related references to discuss the impact and scientific significance of this research and discussed future work related to this study specifically in lines 409-427.

---

## [Editor Report · Decision Letter 2]

18 Nov 2020

Study of the thermal regime of a reservoir on the Qinghai-Tibet Plateau, China

PONE-D-20-19898R2

Dear Dr. Deng,

We’re pleased to inform you that your manuscript has been judged scientifically suitable for publication and will be formally accepted for publication once it meets all outstanding technical requirements.

Kind regards,

NING Sun

Academic Editor

PLOS ONE
---

## [Editor Report · Acceptance letter]

23 Nov 2020

PONE-D-20-19898R2 

·Study of the thermal regime of a reservoir on the Qinghai-Tibetan Plateau, China 

Dear Dr. Deng:

I'm pleased to inform you that your manuscript has been deemed suitable for publication in PLOS ONE. Congratulations! Your manuscript is now with our production department. 

Kind regards, 

on behalf of

Dr. NING Sun 

Academic Editor

PLOS ONE